# Cellular organization in lab-evolved and extant multicellular species obeys a maximum entropy law

**Thomas C Day[1]\*, Stephanie S Höhn[2]\*, Seyed A Zamani-Dahaj[1,3,4], David Yanni[1], Anthony Burnetti[3], Jennifer Pentz[3,5], Aurelia R Honerkamp-Smith[2†], Hugo Wioland[2‡], Hannah R Sleath[2§], William C Ratcliff[3]\*, Raymond E Goldstein[2]\*, Peter J Yunker[1]\***

[1]School of Physics, Georgia Institute of Technology, Atlanta, United States; [2]Department of Applied Mathematics and Theoretical Physics, Centre for Mathematical Sciences, University of Cambridge, Cambridge, United Kingdom; [3]School of Biological Sciences, Georgia Institute of Technology, Atlanta, United States; [4]Quantitative Biosciences Graduate Program, Georgia Institute of Technology, Atlanta, United States; [5]Department of Molecular Biology, Umeå University, Umeå, Sweden

**\*For correspondence:**
tday31@gatech.edu (TCD);
sh753@cam.ac.uk (SSH);
william.ratcliff@biology.gatech.edu (WCR);
R.E.Goldstein@damtp.cam.ac.uk (REG);
peter.yunker@gatech.edu (PJY)

**Present address:** [†]Department of Physics, Lehigh University, Bethlehem, United States; [‡]Institut Jacques Monod de Paris Diderot/CNRS, Paris, France; [§]Department of Chemistry, Imperial College London, London, United Kingdom

**Abstract** The prevalence of multicellular organisms is due in part to their ability to form complex structures. How cells pack in these structures is a fundamental biophysical issue, underlying their functional properties. However, much remains unknown about how cell packing geometries arise, and how they are affected by random noise during growth - especially absent developmental programs. Here, we quantify the statistics of cellular neighborhoods of two different multicellular eukaryotes: lab-evolved 'snowflake' yeast and the green alga *Volvox carteri*. We find that despite large differences in cellular organization, the free space associated with individual cells in both organisms closely fits a modified gamma distribution, consistent with maximum entropy predictions originally developed for granular materials. This 'entropic' cellular packing ensures a degree of predictability despite noise, facilitating parent-offspring fidelity even in the absence of developmental regulation. Together with simulations of diverse growth morphologies, these results suggest that gamma-distributed cell neighborhood sizes are a general feature of multicellularity, arising from conserved statistics of cellular packing.

## Editor's evaluation

This work uncovers a simple but far-reaching statistical principle that describes the geometry of cell packing in snowflake yeast and green algae. It draws on ideas from granular physics to offer new insight into universal rules of multicellular geometry that are otherwise easily obscured by the cell-scale idiosyncrasies of the different biological systems.

## Introduction

The evolution of multicellularity was transformative for life on Earth, occurring in at least 25 separate lineages (*Grosberg and Strathmann, 2007*). The success of multicellular organisms is due in part to their ability to assemble cells into complex, functional arrangements. Self-assembly, however, is fundamentally subject to random noise (*Zeravcic and Brenner, 2014*; *Szavits-Nossan et al., 2014*; *Damavandi and Lubensky, 2019*) that affects the final emergent structure (*Michel and Yunker, 2019*).

The physiology of multicellular organisms can depend sensitively on the geometry of cellular packing (*Bi et al., 2015b*; *Drescher et al., 2016*; *Jacobeen et al., 2018b*; *Brunet et al., 2019*; *Schmieder et al., 2021*), and such noise may therefore have direct consequences on organismal fitness. Understanding the evolution of multicellularity, and the subsequent evolution of multicellular complexity (*Bell and Mooers, 1997*), requires understanding the impact of random noise on multicellular self-assembly. How do organisms accurately assemble functional multicellular components in the presence of noise?

Recent work has shown that extant multicellular organisms can either suppress (*Hong et al., 2016*) or leverage (*Haas et al., 2018*) variability in the process of reliably generating structures, and their tissues can change function based on cellular packing geometry (*Bi et al., 2015a*). This occurs through a coordinated developmental process involving genetic (*Davidson, 2001*), chemical (*Sampathkumar, 2020*), mechanical (*Deneke and Di Talia, 2018*), and bioelectric (*Levin, 2004*) feedbacks between interacting cells. However, even with coordinated developmental processes, noise during self-assembly results in deviations from perfectly regular structures. Further, as these developmental processes have not yet evolved in nascent multicellular organisms, it is unclear how unregulated assembly can reliably result in reproducible packing geometries and multicellular structures.

Multicellular organisms also exhibit diverse growth morphologies; for example, cells can remain attached through incomplete cytokinesis (*Bonner, 1998*; *Grosberg and Strathmann, 2007*; *Knoll, 2011*), they can adhere through aggregative bonds (*Claessen et al., 2014*), and they can assemble multicellular groups through successive cell division within a confining membrane (*Angert, 2005*; *Herron et al., 2019*). These growth morphologies can have distinct intercellular connection topologies (*Yanni et al., 2020*), changing how randomness is manifested. For instance, groups that grow with persistent mother-daughter bonds maintain the same intercellular connections, 'freezing' in place any structural randomness that arises during reproduction. In contrast, cells in aggregates can rearrange, so their final structure emerges from a combination of reproduction and intercellular interactions and noise (*Delarue et al., 2016*; *Hartmann et al., 2019*). Further, the dimensionality of multicellular groups can vary, from filaments (*Herrero et al., 2016*) and quasi-two-dimensional sheets (*Besson and Dumais, 2011*; *Brunet et al., 2019*) to groups that grow equally in three dimensions (*Ratcliff et al., 2012*; *Tang et al., 2020*; *Butterfield, 2016*). While the impact of noise on systems in thermal equilibrium is well known to depend sensitively on spatial dimensionality (*Mermin and Wagner, 1966*; *Hohenberg, 1967*; *Vivek et al., 2017*), no such information is yet at hand for biological development, which is intrinsically out of equilibrium. The growth morphology, connection topology, and dimensionality therefore altogether determine a multicellular architecture. Randomness resulting from many sources, such as stochastic cell division, variability in cell growth, intercellular interactions, and more, subsequently occurs as perturbations to this idealized form. It would appear that noise manifests in a unique, context-dependent manner in each of these different multicellular systems.

Here, we provide experimental evidence that, rather than being context-dependent, fluctuations in cell packing geometry instead follow a universal distribution, independent of the presence or absence of developmental regulation. We quantify the distributions of cellular space in two different types of organisms: experimentally evolved multicellular yeast (*Ratcliff et al., 2012*) and wild-type multicellular green algae (*Goldstein, 2015*). In both cases, maximum entropy considerations (*Aste and Di Matteo, 2008*) (see inset box) accurately predict the cell packing distribution. Building on these observations, we use computational models of diverse prescribed growth rules, mimicking extant biological morphologies, to show that cells are ubiquitously packed according to the maximum entropy principle. Detailed analysis of the case of green algae shows that correlations, that is, the lack of structural randomness, produce deviations from maximum entropy predictions, but that even a relatively small amount of randomness is sufficient to generate cellular packings that largely follow maximum entropy predictions. Next, we explore the evolutionary consequences of cell packing. We use the cell packing distribution to predict the distribution of snowflake yeast group sizes, an emergent multicellular trait that arises from cell crowding (*Jacobeen et al., 2018b*). Then, we use a theoretical analysis to show that the effects of fluctuations in intercellular space on the motility of green algae are small. These findings together suggest that, rather than impeding innovation, fluctuations in cell packing are highly repeatable, and may play a fundamental role in the origin and subsequent evolution of multicellular organisms.

# Results

## Maximum entropy

Within statistical physics, the maximum entropy principle relates randomness in low-level units (e.g. cells) to the properties of the assembly (e.g. a multicellular group). It works by enumerating all low-level configurations that conform to a set of constraints. Any particular group-level property can be generated by many different low-level configurations, but some group-level properties may correspond to more low-level configurations than others. Those that are generated by many configurations are more likely to be observed than those that correspond to relatively few configurations; in this way, the maximum entropy principle allows one to calculate the probability of observing different group properties, given a set of constraints. Multicellular groups obey a simple but universal constraint: each group has some total volume, $V$. This volume can be divided into $N$ pieces, where $N$ is the total number of cells. Each piece is associated with a particular cell, and the $N$ pieces must sum to the total volume of the group, $V = \sum_i v_i$, for $i = 1, 2, ..., N$. Using this constraint, and assuming no correlations, one can predict the most likely distribution of volumes for the $N$ pieces. This approach has been successfully used to predict the distribution of free volumes within granular materials and foams (**Aste and Di Matteo, 2008**; **Katgert and van Hecke, 2010**). Here, we use it to predict the distribution of cellular free volumes in the absence of spatial correlations in cell positions.

Consider the ensemble of all possible cellular configurations in a simple group. As first derived by **Aste and Di Matteo, 2008** and **Aste et al., 2007** for granular materials, the maximum entropy probability distribution $p(v)$ of cell neighborhood volumes within $V$ is the modified gamma distribution

$$p(v) = \frac{k^k}{\Gamma(k)} \frac{(v-v_c)^{k-1}}{(\bar{v}-v_c)^k} \exp\left(-k\frac{v-v_c}{\bar{v}-v_c}\right) \tag{1}$$

where $\bar{v}$ is the mean cell neighborhood volume, $v_c$ is the minimum cell neighborhood volume, $\Gamma(k)$ is the gamma function, and $k \equiv (\bar{v} - v_c)^2/\sigma_v^2$ is a shape parameter that is defined by $v_c$, $\bar{v}$, and the variance of the cell neighborhood volumes, $\sigma_v^2$. This distribution is expected if cell neighborhood volumes are determined independently of each other (while still conforming to the total volume constraint). In other words, volumes must be set randomly; correlations between the size of separate volumes will lead to deviations from maximum entropy predictions. If this condition holds, then maximum entropy volume distribution predictions should be valid, regardless of other geometric or structural details. For example, maximum entropy statistics hold in granular materials, despite the fact that they must obey strict force and torque balance conditions (**Aste and Di Matteo, 2008**; **Snoeijer et al., 2004**; **Bi et al., 2015a**). Further, the same approach applies to groups with a constraint on total area or length; this does not change the result, and $V$ can be replaced by $A$ or $L$ without other modifications. However, we would not expect this prediction to necessarily hold for confluent cell layers (i.e. packing fraction $\phi \approx 1$), since in general cell size is a highly regulated process; when cells tile space, the cell neighborhood volume distribution will strongly correlate with the the cell size distribution.

In practice, we divide the total group volume or area into $N$ pieces via a Voronoi tessellation. The size of the space associated with cell   includes the cell itself and the portion of intercellular space closer to its center than to the center of any other cell. As cells must have non-zero size, we therefore set $v_c$ to be the volume of a single cell without any intercellular space (or $a_c$, the area of a single cell).

## Experimental tests of multicellular maximum entropy predictions

To test whether different kinds of multicellular groups pack their cells according to the maximum entropy principle, we investigated cell packing in two different multicellular organisms. First, we used experimentally evolved 'snowflake' yeast (**Ratcliff et al., 2012**), a model system of undifferentiated multicellularity. Second, we used the green microalga *Volvox carteri*, a member of the volvocine algae that first evolved multicellularity in the Triassic (**Starr, 1969**; **Herron et al., 2009**).

### Snowflake yeast

Snowflake yeast grow via incomplete cytokinesis, generating branched structures in which mother-daughter cells remain attached by permanently bonded cell walls (**Figure 1A**). New buds appear on ellipsoidal cells at a polar angle $\langle\theta\rangle = 42\% \pm 23\%$ and azimuthal angle $\phi$ that is randomly distributed ($\langle\phi\rangle = 180\% \pm 104\%$, **Figure 1—figure supplement 1**). Therefore, even though snowflake yeast structure is limited to configurations in which daughter cells remain attached at their base to their mother,

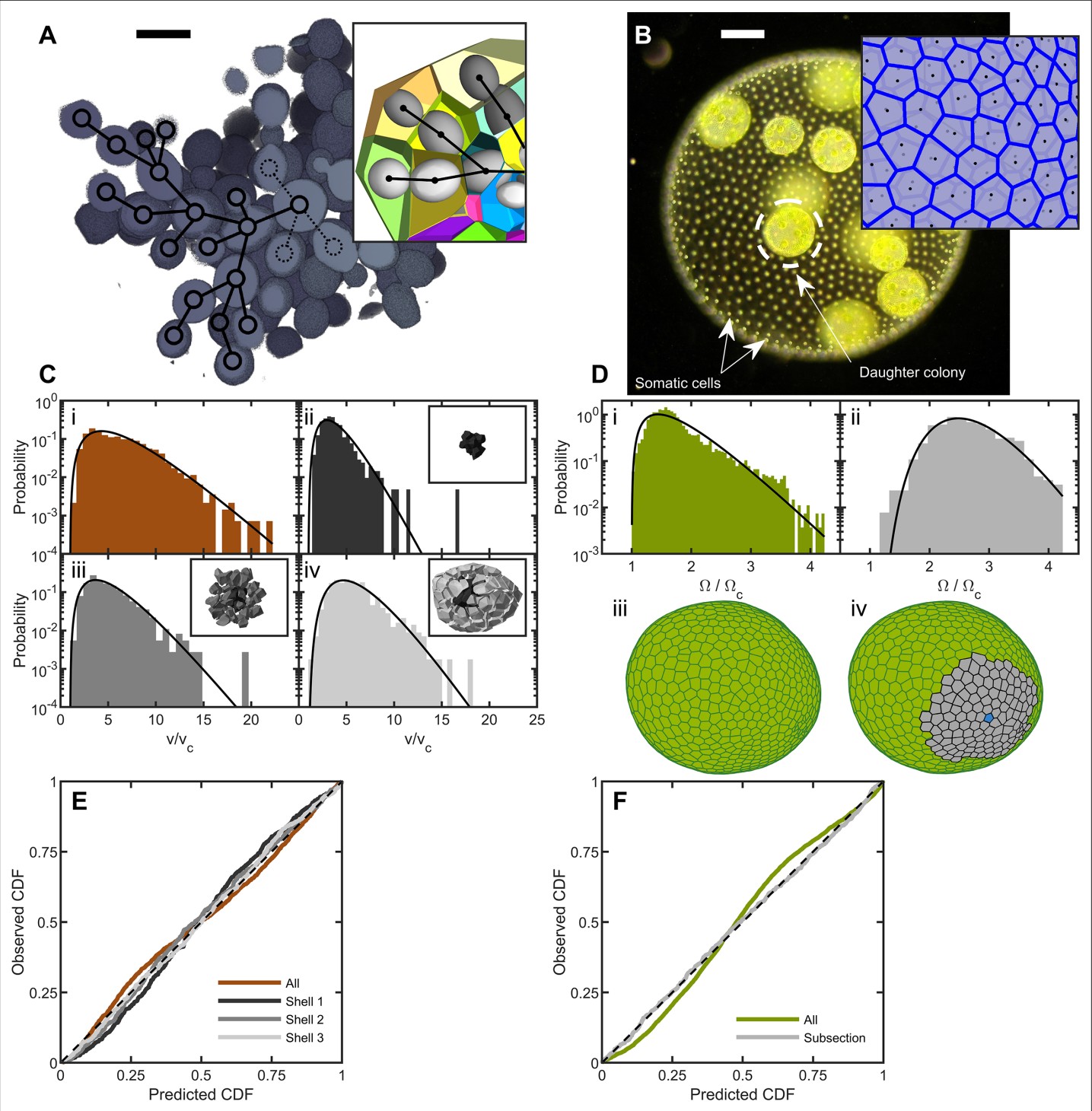

**Figure 1.** Cell packing in two multicellular species. (**A**) Cross-section of a multicellular yeast organism, which grows with persistent intercellular bonds. Scalebar is 5 m. The inset shows a smaller section, with ellipsoidal fits to individual cells along with their corresponding Voronoi polyhedra. Black overlays indicate the connection topology between yeast cells; not all connections are labeled. (**B**) Darkfield microscopy image of *Volvox carteri*, scalebar is $100\,\mathrm{m}$. Inset: a small piece of the Voronoi-tessellated surface; black points are somatic cell positions. (**C**) Distributions of Voronoi polyhedron volumes as a function of cell size normalized by average size $v_c$ for snowflake yeast. In orange is the histogram for all cells; the other three distributions correspond to different subsections of Voronoi volumes. The cells were grouped into spherical shells with radius $R$ and width $\Delta R$ from the cluster center of mass. Shown are shells with edges $[0, 6.2]$, $[6.2, 9.7]$, and $[9.7, 20.4]$. Black lines are maximum entropy predictions. (**D**) Distributions of solid angles subtended by *Volvox* somatic cells divided by a minimum solid angle $\Omega_c$. Solid black lines are the maximum entropy predictions. The top row shows the histogram for all cells in green and a subsection of correlated areas in gray. Bottom row illustrates the subsectioning process: blue polygon is the center

*Figure 1 continued on next page*

*Figure 1 continued*
of the subsectioned region. Only the Voronoi polygons, *i.e.* not the somatic cells, are shown for clarity. (**E,F**) Empirical cumulative distribution function vs entropic predictions for all distributions shown in c,d. The dashed black line represents hypothetical perfect agreement between observation and prediction.

The online version of this article includes the following source data and figure supplement(s) for figure 1:

**Source data 1.** Experimental data files enumerating the cell centers positions for each organism sampled.

**Figure supplement 1.** Random cell budding positions in multicellular yeast groups.

these new cells bud in random orientations. Each snowflake yeast cluster has a different specific configuration of cells; the ensemble of snowflake clusters will include many different configurations. We expect that this structural randomness produces predictable distributions of cellular neighborhood volumes. Conversely, if there are strong correlations in the locations of daughter cells, then we will observe deviations from maximum entropy predictions regarding the cell neighborhood volumes.

To determine the distribution of cell neighborhood volumes, we first must measure the position of every cell in a cluster. It is difficult to image individual cells within snowflake yeast clusters due to excessive light scattering. Instead, we used a serial block face scanning electron microscope equipped with a microtome to scan and shave thin (50 nm) layers off a resin block with embedded yeast clusters with stained cytoplasms. This process allowed us to determine the 3D structure of $N = 20$ snowflake yeast clusters and locate cell centers with nanometer precision.

We define the group volume as the smallest convex hull that surrounds all cells in the cluster and computed the 3D Voronoi tessellation of cell centers within that (**Figure 1A**). The distribution of cellular Voronoi volumes closely matched the predicted k-gamma distribution (**Figure 1C**, $k = 2.88$). This agreement is quantified via 'P-P plots' of the empirical cumulative distribution function (CDF) plotted against the predicted k-gamma CDF. We find a root-mean-square residual $r_{RMS} = \sqrt{\langle (F(v) - F_i)^2 \rangle} = 0.02$, where $F_i$ is the empirical CDF and $F(v)$ is the predicted k-gamma CDF.

The influence of the convex hull on these results was investigated by using an alternative procedure in which the Voronoi volumes were binned into shells centered at the cluster's center of mass (**Figure 1C and E**). We binned cells into shells with shell edges of $[0, 6.2]$, $[6.2, 9.7]$, and $[9.7, 20.4]$ away from the center of mass. We found that the distribution of Voronoi volumes within each shell matched the predicted k-gamma distribution, with $r_{RMS} = \{0.037, 0.020, 0.014\}$, $k = \{3.45, 3.08, 4.63\}$ in the shells shown in **Figure 1C(ii-iv)**.

## Volvocine algae

To test if cell neighborhood volumes in extant multicellular organisms are consistent with maximum entropy cell packing predictions, we examined cell packing within the green microalgae *Volvox carteri*. Development in *V. carteri*, which evolved over millions of years, is highly regulated, occurring through a stereotyped morphological progression (**Kirk, 2005**). *V. carteri* embryos arise as a spherical cellular monolayer from palintomic cell divisions with incomplete cytokinesis, which leaves the cells attached via cytoplasmic bridges. These bridges disappear when ECM is secreted by the cells, filling the entire sphere, and eventually moving the cells apart. The approximately 1000 somatic cells remain embedded on the surface of a translucent sphere of extracellular matrix (**Figure 1B**). While six-fold coordination is the most frequent local arrangement of somatic cells, the fact that the cells are embedded in a surface with spherical topology requires there to be 'defects' with differing coordination number (e.g. 5,7), and these are found interspersed around the spheroid. Thus, despite their developmental regulation, somatic cells exhibit a degree of disorder with respect to coordination number. From a physics perspective, the local hexatic order in the somatic cell arrangement is low (see Materials and methods).

To determine the distribution of *Volvox* cell neighborhood sizes, we imaged somatic cells using their chlorophyll autofluorescence in a light sheet microscope. Since the somatic cells are arranged around a surface embedded in 3D space, we constructed a 2D Voronoi tessellation of somatic cells on the surface. Each organism imaged had a different size, and therefore had a different mean Voronoi area $\langle A \rangle$. To compare distributions across organisms, we removed the systematic area differences by recording the solid angle $\Omega_i = 4\pi A_i / S$ subtended by each somatic cell, where $S = \sum_i A_i$ is the total surface area of the organism. We found that the k-gamma distribution largely matched the distribution

of solid angles (*Figure 1D*, $k = 2.40$, $r_{RMS} = 0.04$). However, there are systematic deviations between the data and maximum entropy predictions (*Figure 1F*).

We next investigated if maximum entropy predictions are more accurate within subregions with similar mean solid angles; specifically we examine regions whose mean is $\langle \Omega \rangle = 0.0185 \pm 0.0003$, obtained across six organisms. The distribution of Voronoi solid angles within these subregions closely follows the k-gamma distribution (*Figure 1F*, $k = 10.66$, $r_{RMS} = 0.01$). This observation suggests that while there are systematically correlated subregions of cells, within these subregions cells are largely arranged randomly. Thus, the organization of *Volvox carteri* somatic cells is consistent with maximum entropy predictions.

## Simulations of different growth morphologies

We next used simulations to investigate the impact on cell packing of four different growth morphologies: growth via incomplete cell division (*cf.* snowflake yeast), cells distributed on a spherical surface (*cf. Volvox*), aggregation, and palintomy. The goal of these studies was to determine if morphological details and constraints impact entropic packing using simplified models that capture the essential features of the growth and behavior of these varied organisms. These simulations captured a couple basic properties of multicellularity: organisms may grow in both two and three dimensions, and they may assemble with two different classes of bonds: bonds that are reformable, and bonds that are not reformable. Nonreformable, or 'fixed' bonds (like those formed from incomplete cytokinesis) are common in plants, algaes, and fungi; on the other hand, reformable bonds (such as those formed via sticky proteins) are common in animal tissues and cellular aggregates, like biofilms and slime molds. Finally, palintomy, or growth confined inside a maternal membrane, is also common from algae to animals.

First, we performed geometric simulations of multicellular groups that grow via incomplete cell division; these simulations were inspired by previous simulations of snowflake yeast (*Jacobeen et al., 2018a*; *Jacobeen et al., 2018b*). Daughter cells bud from mother cells with experimentally determined polar angle and random azimuthal angle, and remain attached to mother cells with rigid bonds. We ran 9, 100 simulations starting from a single cell, each of which underwent seven generations of division, and calculated the Voronoi tessellation of the final structure from each simulation. The distribution of Voronoi volumes closely matched the k-gamma distribution across four orders of magnitude (*Figure 2A*, $k = 2.26$, $r_{RMS} = 0.007$).

Inspired by *Volvox*, we simulated cells distributed across the two-dimensional surface of a sphere through a random Poisson point process. We completed 10 simulations, each with 1000 cells, and computed the distribution of solid angles subtended by Voronoi cells. As shown in *Figure 2B*, the distribution of Voronoi solid angles is consistent with maximum entropy predictions ($k = 9.29$, $r_{RMS} = 0.009$).

Next, we simulated organisms that stick together via reformable cell-cell adhesions, a mechanism of group formation that is common in biofilms and extant aggregative multicellular life (*Claessen et al., 2014*) (i.e. *Dictylostelium* and *Myxococcus*; *Figure 2C*). In these simulations, multicellular aggregates were grown from a single cell. Seven generations of cell division occurred, in which new cells appear on the surface of existing cells at random positions, and steric interactions force cells to separate after division and occupy space. Aggregative bonds were modeled through harmonic interactions of the cell centers. The observed Voronoi volume distributions were consistent with maximum entropy predictions ($k = 7.84$ and $r_{RMS} = 0.007$).

Finally, we modeled cells undergoing palintomic division within a maternal cell wall, as is common in green algae (*Lürling and Van Donk, 1997*; *Boraas et al., 1998*; *Ratcliff et al., 2013*; *Fisher et al., 2016*; *Herron et al., 2019*), and is reminiscent of both baeocyte production in *Stanieria* bacteria (*Angert, 2005*) and neoproterozoic fossils of early multicellularity (*Xiao et al., 1998*; *Figure 2D*). The details of these simulations remained similar to the simulations of aggregative multicellularity, with the important difference being that instead of harmonic interactions between cell centers enforcing groups to stay together, cells interacted with a spherical maternal wall acting as a corral. The Voronoi volume distributions for these simulations were also consistent with maximum entropy predictions ($k = 15.16$ and $r_{RMS} = 0.013$).

Taken together, the results of these simulations suggest that a broad distribution of cell neighborhood sizes is a general feature of multicellular growth morphologies. In particular, when cell locations

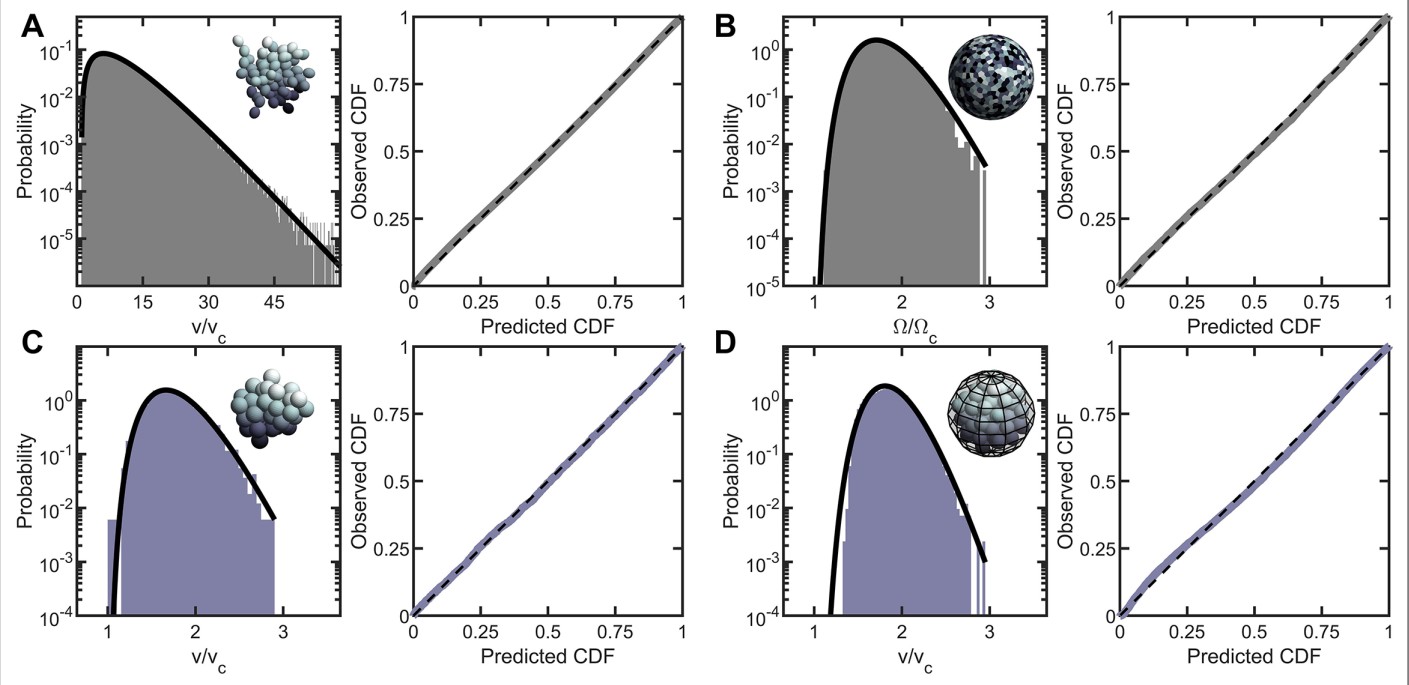

**Figure 2.** Entropic packing is a general feature of simple multicellularity. We simulated four different growth morphologies: (**A**) Tree-like groups formed with rigid, permanent bonds between cells, (**B**) surface-attached cells located on a spherical surface, (**C**) aggregates formed with attractive 'sticky' interactions, and (**D**) groups formed by rapid cell division within a maternal membrane. In all subfigures, left panel shows the predicted and observed probability distributions, and right panel plots the observed cumulative distribution vs. the expected cumulative distribution. Histogram bars represent measured Voronoi volume distribution in simulations, and black solid line represents the maximum entropy prediction. Maximum entropy predictions accurately described the distribution of cellular volumes/areas, despite their varying mechanisms of group formation ($r_{RMS} \leq 0.01$).

The online version of this article includes the following source data and figure supplement(s) for figure 2:

**Source data 1.** Simulation data (enumerating the cell center positions) for the six classes of numerical studies; aggregation, apoptosis, polydispersity, snowflake yeast growth, tree-like growth, and Volvox growth.

**Figure supplement 1.** Four different distributions were tested for goodness-of-fit: the maximum entropy prediction (black line), the normal distribution (red), the log-normal distribution (blue), and the beta-prime distribution (green).

**Figure supplement 2.** Four different distributions were used to estimate the skewness of the distribution given the first two moments (**A**).

**Figure supplement 3.** In simulations of palintomy within a confining membrane, boundary effects due to frustrated packing near a wall affect the local packing fraction, but do not change the general distribution.

are random under these rules, cell neighborhood size distributions closely follow the k-gamma distribution.

## The role of spatial correlations

While we have shown that the distribution of cell neighborhood volumes closely follows the k-gamma distribution in two very different organisms, we have also seen that in some cases maximum entropy predictions are more accurate in sub-sections of an organism than across its entirety. For instance, in *Volvox* we observed that $r_{RMS}$ is much smaller within subregions with similar mean solid angles than across the whole organism. This observation suggests that correlations exist in the arrangement of *V. carteri* somatic cells. Such a correlation is not wholly unexpected; *V. carteri* have a well-known anterior-posterior polarity which they use to orient and swim toward light sources (*Drescher et al., 2010b*). This polarity may affect the somatic cell packing distribution in different subregions of the organism, leading to deviations from maximum entropy predictions.

The spatial correlations in the cellular areas in *Volvox* were studied first by plotting a 3D heatmap of Voronoi solid angle sizes (*Figure 3A*). It is apparent that extended spatial regions have well-defined and non-random mean Voronoi solid angles. We quantified this feature by calculating the spatial correlation function $C(Q)$ of the solid angle

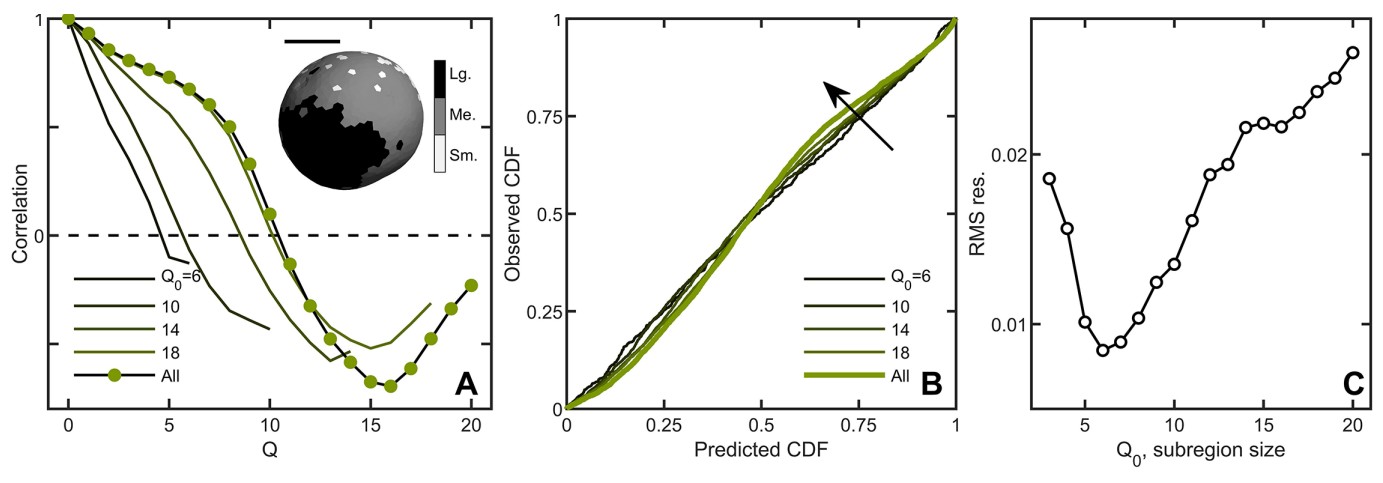

**Figure 3.** Correlations lead to deviations from maximum entropy predictions in *Volvox carteri*. (**A**) Correlation function of Voronoi polygon areas vs. network neighbor distance $Q$. Green circles represent all experimental *Volvox* data. Lines indicate the same correlation function calculated in subsections of size $Q_0 = \{6, 10, 14, 18\}$. Inset: visualization of spatial correlations of solid angle; one *Volvox*'s Voronoi tessellation is displayed with a three-color heatmap corresponding to polygons with areas smaller than (light gray), within (gray) and larger than (black) than one standard deviation of the mean. Scale bar is $200\,\text{m}$. (**B**) PP plots for the observed vs predicted cumulative distribution function. In green is the *Volvox* distribution for all cells before corrections for correlations. A selection of differently-sized subsections is also plotted, corresponding to sizes $Q_0 = \{6, 10, 14, 18\}$. Arrow indicates direction of increasing $Q_0$ value. (**C**) Root-mean-square residual deviation from maximum entropy predictions as a function of subsection size, as a function of nearest neighbor number $Q_0$. As the subsection size increases (including more and more uncorrelated Voronoi areas), the deviation from predictions first decreases until $Q_0 = 6$, then increases.

The online version of this article includes the following source data for figure 3:

**Source data 1.** Used in this figure is specifically Volvox cartesian cell center position measured from lightsheet microscopy.

$$C(Q) = \frac{\langle (\Omega - \langle\Omega\rangle) Y_Q \rangle}{\sigma_\Omega \sigma_{Y_Q}}, \qquad (2)$$

where $Y_Q = J(Q)^{-1} \sum_j (\Omega_j - \langle\Omega\rangle)$ is the average deviation of the solid angle of a given polygon's neighbors at a neighbor distance $Q$ from the mean. Here, the number of neighbors is $J(Q)$, a function of $Q$, which enumerates the network distance from the polygon of interest (i.e. $Q = 1$ calls the nearest neighbors, of which there are $J(1)$, $Q = 2$ calls the next nearest neighbors, of which there are $J(2)$, and so on). The standard deviation of the solid angle across the population is $\sigma_\Omega$, and $\sigma_{Y_Q}$ is the standard deviation of $Y_Q$ across the population. We find that *Volvox* Voronoi solid angles are positively correlated at distances as large as $Q = 10$ (**Figure 3A**). This analysis suggests that there are systematic differences in *Volvox* group structure in different spatial regions. We therefore should expect to observe deviations from the k-gamma distribution, which was derived under the assumption that there are no correlations in the division of space among cells.

A natural question is whether maximum entropy predictions are more accurate within correlated subregions of an organism. We measured the Voronoi distribution in subregions with similar mean solid angles across six organisms and, for each subregion, a central node and its neighbors up to $Q_0$ were identified. We varied $Q_0$ from 3 (corresponding to, on average, $38 \pm 2$ cells in the subregion) to $Q_0 = 20$ ($928 \pm 10$ cells on average in the subregion) to measure the Voronoi solid angle distributions in subregions of different sizes. Spatial correlations were weaker within smaller domains (**Figure 3A**), and deviations from maximum entropy predictions smaller as well (**Figure 3B**) with the minimum $r_{RMS}$ at $Q_0 = 6$ (**Figure 3C**, average of $133 \pm 5$ cells). These observations suggest that while there are systematic correlations between subregions, cell neighborhood sizes are largely randomly distributed within subregions.

## The crucial role of randomness

How much randomness is necessary for the k-gamma distribution to predict cell neighborhood size distributions? Our analysis of the solid angle distribution of *Volvox* cells demonstrates that maximum entropy principle predictions are relatively accurate ($r_{RMS} = 0.04$) even in the presence of some spatial

correlations. However, it is unclear how much randomness is necessary for the k-gamma distribution to be predictive. Developmental processes are known to incur random effects (*Hong et al., 2016*; *Haas et al., 2018*), but, at the same time, fluctuations must be sufficiently suppressed for complex structures to assemble. This dilemma is augmented because there are few methods for measuring the degree of randomness in developmental processes. Therefore, it is difficult to disentangle exactly which reproducible qualities and traits stem from regulated processes, and which ones stem from random processes. In this section, we address this gap by assessing the stability of maximum entropy packing distributions to correlative perturbations (*Figure 4*). We do so by simulating three different sources of correlations: (i) size polydispersity, (ii) precisely defined growth patterns, and (iii) coordinated cellular apoptosis, each of which is a common multicellular developmental process. In each scenario, we varied the relative strength of correlations and noise, and monitored how closely the cell neighborhood size distributions agreed with the k-gamma distribution via P-P plots and the $r_{RMS}$. We do not attempt to exactly recapitulate naturally-occurring levels of randomness strength; rather, we investigate the limits of how well maximum entropy predictions can measure the impact of correlations.

The impact of heritable size polydispersity was investigated by simulating aggregative groups consisting of large and small cells. All simulations were seeded with one small cell and one large cell. We then varied the probability $\xi$ that a new cell is the same size as its mother from 0.5 to 1.0. When $\xi = 1$, cells always produce offspring with the same radius; for $\xi = 0.5$ it is equally likely that a small cell produces a small or large daughter (and vice versa for large cells). Therefore, groups with $\xi = 1$ have correlated regions of cell size, but the degree of correlations decreases with decreasing $\xi$. While groups with $\xi = 1$ deviate significantly from the k-gamma distribution ($r_{RMS} = 0.09$), we observed that even a small amount of randomness results in excellent agreement between simulated groups and the k-gamma distribution (in order from $\xi = 1$ to $\xi = 0.5$, $r_{RMS} = \{0.09, 0.04, 0.03, 0.03, 0.02\}$).

Next we investigated groups with varying amounts of noise on top of defined growth patterns. In these simulations, new cells bud in precise positions; the first daughter at the position ($\theta = 0 \pm \eta$, $\phi = 0 \pm \eta$ in spherical coordinates), the second at $\theta = 90 \pm \eta$, $\phi = 0 \pm \eta$, the third at $\theta = 90 \pm \eta$, $\phi = 180 \pm \eta$, etc., where the noise is uniformly distributed with zero mean and width $\eta$. For $\eta = 0$ (no noise), the distribution of Voronoi volumes was discontinuous, since cells could only access a finite number of local configurations. As expected, as $\eta$ increases ($\eta = \{0, 5, 30, 60, 90\}$), $r_{RMS}$ decreases ($r_{RMS} = \{0.07, 0.05, 0.02, 0.02, 0.02\}$).

Finally, we investigated groups with localized and random cell death. In these simulations, 50 cells were confined to the surface of a sphere of unit radius following the protocol described above. One cell is randomly selected to die. Centered at this cell, a spherical region of radius $R$ is defined, and then 10 cells in this region were randomly selected to die (and disappear, thereby not contributing to the Voronoi tessellation). For small $R$, cell death is highly localized, and thus spatially correlated. As $R$ increases, cell death events become less localized, and therefore more random. We find that highly correlated cell death resulted in large deviations from maximum entropy predictions. Conversely, as $R$ increases dead cells become less localized, the observed distribution becomes more accurately described by the k-gamma distribution; as $R$ increases from $R = \{0.75, 1, 1.25, 1.5, 1.75, 2\}$, we find $r_{RMS} = \{0.10, 0.07, 0.03, 0.02, 0.01\}$.

In summary, absent randomness, spatial correlations lead to large deviations from the k-gamma distribution. Yet, with even a small amount of randomness, the k-gamma distribution holds significant predictive power. These simulations suggest that maximum entropy predictions are likely to be robust against even moderate correlations.

## Parent-offspring fidelity via maximum entropy packing

So far, we have shown that randomness in cellular packing leads to highly predictable packing statistics. Here we show that maximum entropy statistics can directly impact the emergence of a highly heritable multicellular trait, organism size.

Prior work has shown that the size of snowflake yeast at fragmentation is remarkably heritable – higher, in fact, than the traits of most clonally reproducing animals (*Ratcliff et al., 2015*). The size to which snowflake yeast grow depends strongly on the aspect ratio of its constituent cells; more elongated cells allow the growth of larger clusters before strain from cellular packing causes group fragmentation (*Jacobeen et al., 2018a*; *Jacobeen et al., 2018b*). Recently, experiments with engineered yeast showed that this emergent multicellular trait, group size, was in fact more heritable than the underlying

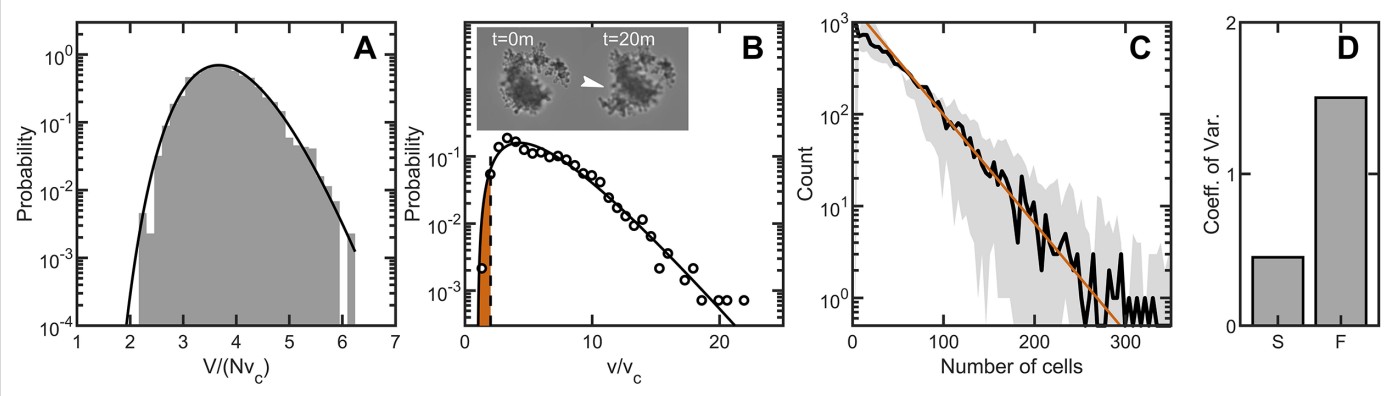

**Figure 5.** Maximum entropy cell packing generates a consistent and predictable life cycle in snowflake yeast. (**A**) Distribution of total cluster volume for 3000 simulated snowflake clusters, each with $N = 100$ cells. The total volume is divided by the minimum possible volume $Nv_c$. The k-gamma distribution (black line, $k = 23.0$, $r_{RMS} = 0.0043$) provides a good description of the data. (**B**) Distribution of all experimental Voronoi volumes (black circles) and the maximum entropy prediction (black line). The vertical black dashed line is the critical Voronoi volume $v^* = 2.02$ predicted from simulations. Orange filled region integrates up to $p^*$ the probability that any one cell occupies a volume less than $v^*$. Insets: sequential brightfield microscope images of one yeast cluster undergoing group fragmentation. White arrowhead indicates location of fracture point. The images measure 150 m across from top to bottom. (**C**), Experimentally measured yeast cluster size distribution (solid black line) along with the prediction from weakest link theory (orange line). Gray region represents $1\sigma$ confidence bounds on the measured distribution from estimating the number of cells in a group. (**D**), Coefficient of variation in group radius ($\sigma/\bar{R}$) for snowflake groups (**S**) and flocculating groups (**F**). Data from *Pentz et al., 2020*.

The online version of this article includes the following source data for figure 5:

**Source data 1.** This figure was generated from a combination of the simulation data and experimental data provided with *Figures 1 and 2*, as well as epxerimental size measurements included here from a particle multisizer, or Coulter Counter.

cellular trait upon which it was based (cellular aspect ratio), despite the fact that the mutations engineered in this system only affected cellular aspect ratio directly (*Zamani-Dahaj et al., 2021*). Simulations of multicellular chemotaxis observed a similar effect (*Colizzi et al., 2020*). While at first glance this may seem surprising, we show below that the high heritability of snowflake yeast group size arises from the direct dependence of size on the robust maximum entropy distribution of volume within groups.

Before addressing how fracture impacts the distribution of cluster sizes by impacting the number of cells within a group, we first must address fluctuations in size among clusters with the same number of cells. Given a number of cells $N$ in the cluster, variation in cell packing fraction results in variation of the total volume. The arguments given above for predicting the distribution of individual cell volumes also applies to the distribution of total volume (*Aste and Di Matteo, 2008*); the distribution of total volume for clusters with the same number of cells should follow the k-gamma distribution. To generate enough clusters with identical $N$ to test this prediction, we used simulations. We generated 3000 snowflake yeast clusters, each with 100 cells, and measured their total volumes. The distribution of volumes is consistent ($r_{RMS} = 0.0043$, $k = 23.0$) with the k-gamma distribution as shown in *Figure 5A*. Further, these fluctuations in size are small compared to the differences in size gained via reproduction of cells or lost via fracture.

To predict the group size distribution, we consider the probability of fragmentation via a weakest-link model of fracture. As the location of new cells is random (see *Figure 1—figure supplement 1*), each new cell has a chance of causing intercellular bond fracture. It was previously observed that bonds only break if cells are highly confined, that is they have smaller Voronoi volumes; otherwise flexible cellular branches simply bend (*Jacobeen et al., 2018b*). We model fracture as occurring when a cell's Voronoi volume is below a critical value denoted by $v^*$ (*Figure 5B*) such that its motion is completely restricted. We measure $v^*$ from simulations that determine the maximum local packing density for groups with same cell size and shape distributions as seen in experiments (see Materials and methods). The probability that a particular cell is confined to a Voronoi volume $v \leq v^*$ is the integral

$$p^* = \int_{v_c}^{v^*} p(v)dv.$$

(3)

As each cell in a cluster of $N$ cells independently has probability $p^*$ of having $v \leq v^*$ (and thus causing fracture), the probability of a cluster with $N$ cells not fragmenting is

$$P(N) = (1 - p^*)^N \tag{4}$$

As we do not model the fate of products of fragmentation (i.e. the size of the separate pieces post-fracture), we expect the weakest link model to be more accurate for larger clusters than it is for smaller clusters.

We measured group size for approximately $10,000$ snowflake clusters, all descendants of a single isolate, using a particle multisizer, and found strong agreement between the experimentally observed cluster size distribution and the weakest-link prediction (the coefficient of determination is $r^2 = 0.97$ for $\log(Counts)$ vs $N$) (*Figure 5C*). Hence, the predictable statistics of entropic cell packing guides the distribution of group size among offspring of a single isolate.

For context, we compared the distribution of group size in snowflake yeast to that of flocculating yeast, which forms multicellular groups via aggregation. The multicellular size of flocculating yeast depends on the rate of collisions with other cells and groups of cells. The growth rate of aggregates is thus typically proportional to their size, as larger aggregates are more likely to contact more cells (*Pentz et al., 2020*). In fact, the maximum size of a flocculating yeast aggregate is bounded by the duration of aggregation, an extrinsic parameter, while the minimum size can be a single cell (*Stratford, 1992*). Using data from *Pentz et al., 2020*, we compared the group size distributions of snowflake yeast and flocculating yeast grown in the same environmental conditions. We find that flocculating yeast groups exhibit a much larger coefficient of variation in size compared to snowflake yeast groups (*Pentz et al., 2020*; *Figure 5D*). These results demonstrate that randomly assembled groups can exhibit more reproducible group traits than groups assembled with correlations.

## Multicellular motility is robust to cellular area heterogeneity

One of the issues arising from the existence of the broad distribution of somatic cell areas in *Volvox* is the extent to which colony motility is affected by that heterogeneity. Each of the somatic cells at the surface of a *Volvox* colony has two flagella that beat at $\sim 30$ Hz, in planes that are primarily oriented in the anterior-posterior (AP) direction but with a slight lateral tilt that makes each colony spin around its AP axis. A longstanding focus in biological fluid mechanics of multicellular flagellates has been to understand the connection between the beating of the carpet of flagella that cover their surface and their self-propulsion. Measurements of the flow fields around micropipette-held (*Short et al., 2006*) and freely-swimming colonies (*Drescher et al., 2010a*) have shown that despite the discreteness of the flagella, the flow is remarkably smooth, albeit often displaying metachronal waves (*Brumley et al., 2015*), long-wavelength phase modulations of the beating pattern. However, the effect of heterogeneity in cell area on motility is as yet unknown; naively, we might expect that heterogeneities in flagella packing may lower the motility of the organism in comparison to a highly regular arrangement. Here, we show in fact that *Volvox* motility is largely unaffected by fluctuations in cell area, indicating that area heterogeneity may not be detrimental to the organism's swimming fitness. The absence of such a barrier may represent a scenario where trait optimization can be achieved without first evolving sophisticated developmental genes.

A heuristic explanation for the smoothness of the flows can be developed by noting first that the flow arising from each flagellum, beating close to the no-slip surface of the colony, will fall off only as an inverse power of distance $r$ from the flagellum. Thus, the superposition of the flows from many flagella will be sensitive to contributions from distant neighbors (*Boselli et al., 2021*) and will tend to wash out local variations in flagellar actuation. This argument can be made quantitative using two different models for the motility of such flagellates. The first is the 'squirmer' model (*Lighthill, 1952*), in which the flagellate is characterized by a tangential 'slip' velocity $u(\theta)$ on the surface, which can be thought of as corresponding to the mean motion of the flagella tips. Here, $\theta \in [0, \pi]$ is the polar angle with respect to the AP axis. In this approach the details of the fluid velocity profile below the tips are not resolved, and in particular the no-slip condition at the surface of the ECM is ignored. In the second approach (*Ishikawa et al., 2020*), which builds on earlier work (*Short et al., 2006*) that specified a force density at the colony surface instead of a slip velocity, there is a specified force density applied at some small distance above the no-slip colony surface, and the flow field below that locus is resolved. This approach, termed the 'shear stress, no-slip' model, captures the very large viscous

dissipation that occurs in the region between the ECM and the locus of forcing. Within either of these two approaches above the effects of area inhomogeneities can be investigated by coarse-graining the flagella dynamics; either the local slip velocity $u(\theta)$ or the local tangential force density $f(\theta)$ has noise.

In the squirmer model, the swimming speed $U$ is (**Lauga, 2020**)

$$U = \frac{1}{2}\int_0^\pi d\theta \sin\theta\, u_\theta(\theta) V_1(\theta), \tag{5}$$

where

$$V_n(\theta) = \frac{2}{n(n+1)} P_n'\left(\cos\theta\right)\sin\theta, \tag{6}$$

$P_n$ is the Legendre polynomial, and the prime indicates differentiation with respect to its argument. If we represent the effects of area inhomogeneities as noise in the slip velocity, then it is most natural to use $V_n$ as the basis functions for the tangential slip velocity, expressed as

$$u_\theta(\theta) = \sum_{n=1}^\infty u_n V_n(\theta), \tag{7}$$

where $V_n(0) = V_n(\pi) = 0$, guaranteeing that the slip velocity vanishes at the anterior and posterior poles (**Short et al., 2006**). Accurate experimental measurements of the azimuthal velocity field of *Volvox* (**Drescher et al., 2010b**) show that it is well-captured by that lowest mode, leading to a modest anterior-posterior asymmetry. From the orthogonality relation for the $V_n$,

$$\int d\theta \sin\theta\, V_1(\theta) V_n(\theta) = \frac{2n(n+1)}{2n+1}\delta_{1n}, \tag{8}$$

we see immediately that the contributions from all modes $n > 1$ vanish identically, and thus the swimming speed is given identically by the amplitude of the lowest mode $V_1(\theta) = \sin\theta$,

$$U = \frac{2}{3}u_1. \tag{9}$$

Thus, within the squirmer model, motility is essentially insensitive to area inhomogeneities. This result does not preclude effects of those higher modes, only that such effects will be on quantities other than the swimming speed, such as the nutrient uptake rate (**Magar et al., 2003**).

In the shear-stress, no-slip model, the velocity field in the region between the colony radius $R$ and the radius $R(1 + \epsilon)$ at which the shear stress is applied is solved separately from that for $r > R(1 + \epsilon)$ and the two flow fields are matched at $R(1 + \epsilon)$ through boundary conditions of continuity in velocity and normal stress and the specified discontinuity in shear stress. Analogously to the expansion of the slip velocity in the squirmer model (7), noise in that discontinuity can be expressed by assuming that the coarse-grained shear force applied by the flagella has spatial variations, and can be expanded in the form

$$f_\theta(\theta) = \sum_{n=1}^\infty f_n V_n(\theta). \tag{10}$$

The swimming speed again depends only on the lowest-order mode in this expansion,

$$U = \frac{2\epsilon R}{3\mu}f_1, \tag{11}$$

and we again have insensitivity of $U$ to inhomogeneities in the area per somatic cell.

## Discussion

In this paper, we demonstrated that universal cellular packing geometries are an inevitable consequence of noisy multicellular assembly. We measured the distribution of Voronoi polytope sizes in both nascent and extant multicellular organisms, and showed that they are consistent with the k-gamma distribution, which arises via maximum entropy considerations. Using simulations, we demonstrated that k-gamma distributions arise in many different growth morphologies, and do so requiring only a relatively small amount of structural randomness. Further, we showed that the distribution of cell neighborhood sizes can be used to distinguish the effects of randomness from the effects of developmental

patterning. Finally, we demonstrated that consistent packing statistics can lead to highly reproducible, and thus heritable, multicellular traits, such as group size in snowflake yeast. Altogether, these results indicate that entropic cell packing is a general organizing feature of multicellularity, applying to multicellular organisms with varying growth morphologies, connection topologies, and dimensionalities.

One of the strengths of the packing-based maximum entropy framework employed here is its simplicity. We have demonstrated that the distribution of cell neighborhood sizes can be predicted with high accuracy, in many different multicellular morphologies, from only the first two moments of the distribution. Deviations from maximum entropy predictions therefore encode important information about additional correlations that can arise via a variety of sources, such as developmental regulation or interactions with the environment. These additional correlations could be explored via, for example, higher order structures to the maximum entropy model (*Schneidman et al., 2003*). Relatedly, it would be interesting to extend our analysis to include topological information, which has recently been applied to theoretical and experimental systems (*Skinner et al., 2021*; *Jeckel et al., 2021*). Developing methods to incorporate these higher-order effects may prove especially useful when assessing the importance of random fluctuations in organisms with structures more complex than those studied here.

The effect of random noise has been an important area of research in developmental biology (*Tsimring, 2014*; *Lander, 2011*). During development, cellular growth, reproduction, differentiation, and patterning combine to form a multicellular organism. Random noise introduced at any stage in this process can result in phenotypic variability, which may affect an organism's fitness (*Waddington, 1957*). But while some multicellular traits exhibit high variability, others are tightly conserved, leading to a wide body of research addressing the origin of mechanisms underlying robustness and stability, and the nature of feedback mechanisms that must be present to manage the large number of stochastic fluctuations in gene expression and growth (*Gregor et al., 2007*; *Haas et al., 2018*; *Hong et al., 2016*; *Sampathkumar, 2020*; *Deneke and Di Talia, 2018*). In this context, our results demonstrate that random noise can itself lead to highly reproducible multicellular traits such as the cell packing distribution.

Our observation that heritable properties can arise from random processes is reminiscent of the reproducible structures and phenomena generated by random noise in a wide range of physical (*Shinbrot and Muzzio, 2001*; *Manoharan, 2015*) and biological systems (*Tsimring, 2014*; *Lander, 2011*). While it may be surprising that the distribution of free space in snowflake yeast and *Volvox* follow the same k-gamma distributions despite the many differences between these organisms, this universality actually extends beyond multicellular organisms to non-living materials, such as those seen in granular materials and foams (*Katgert and van Hecke, 2010*; *Varadan and Solomon, 2002*; *Aste and Di Matteo, 2008*). This broad universality likely arises due to the simple requirements for application of the maximum entropy principle to packing; specifically, there must be a total volume, individual volumes cannot overlap, and volumes must be determined independently (subject to the total volume constraint). It is thus important to note that entropic packing is not necessarily adaptive; it can readily emerge as a consequence of random cellular reproduction or interactions. While entropic packing statistics may produce advantages in some cases, they could be neutral or detrimental in others.

An example of one possible advantage granted by entropic packing is the parent-offspring fidelity that arises from its ensemble statistics. Since both parents and their offspring are assembled through similar noisy processes, they achieve similar cell packing distributions. This statistical similarity therefore details at least one heritable multicellular trait that does not rely on genetically regulated multicellular development. Other multicellular traits that build on the cell packing distribution are similarly affected by this emergent process and could become heritable as well. Such parent-offspring heredity could play a crucial role in the evolutionary transition to multicellularity, providing a mechanism for nascent multicellular organisms to participate in the evolutionary process without first having to possess genetically regulated development. Over time, developmental innovation may arise via multicellular adaptation, modifying or replacing entropic cell packing as a mechanism of multicellular heredity. Consistent with this hypothesis, maximum entropy retains considerable predictive power in extant multicellular organisms such as *Volvox*, animal embryos (*Alsous et al., 2018*), and epithelial tissue monolayers (*Atia et al., 2018*), each of which have canalized development. There may be other examples of organisms with highly regulated development which pack cells according to maximum entropy predictions, and future work could address cell packing in, for example, animal embryos,

brain tissue, and more. Finally, as fragmentation is a common mode of multicellular reproduction (*Larson et al., 2020*; *Prakash et al., 2021*; *Angert, 2005*; *Keim et al., 2004*; *Koyama et al., 1977*), fracture driven by maximum entropy packing statistics may be relevant to organisms other than snowflake yeast.

The broad distributions in cellular volumes we have found in two very different types of organisms, with two very different modes of reproduction and growth, suggest that noise in developmental geometry may be an inevitable consequence of almost any microscopic mechanism. In this sense, they may be just as unavoidable in biological contexts as thermal fluctuations are in systems that obey the rules of equilibrium statistical physics. As an example, we recall the 'flicker phenomenon' of erythrocytes, in which the red blood cell membrane exhibits stochastic motions around its equilibrium biconcave discoid shape. Thought for many years to be a consequence of specific biochemical processes associated with living systems, flickering was eventually shown by quantitative video microscopy (*Brochard and Lennon, 1975*) to be consistent with equilibrium thermal fluctuations of elastic biomembranes immersed in water. This was later confirmed by similar studies of shape fluctuations exhibited by large lipid vesicles (*Schneider et al., 1984*). The generalization of these considerations to homeostatic tissues with cell division, rearrangements and apoptosis has also been considered (*Risler et al., 2015*; *Kalziqi et al., 2018*). While such membrane systems may differ greatly in the specific values of their elastic modulus (and, indeed, of their microscopic membrane constituents), the viscosity of the surrounding fluid, and their physical size, the space-time correlation function of fluctuations about the equilibrium shape adopts a universal form in appropriately rescaled length and frequency variables.

These results on equilibrium fluctuations provide a conceptual precedent for the results reported here. A central issue that then arises from our results is how to connect any given stochastic biochemical growth process defined at the microscopic level to the more macroscopic probability distribution function observed for cellular volumes. Mathematically, this is the same question that arises in the theory of random walks, wherein a Langevin equation defined at the microscopic level leads, through suitable averaging, to a Fokker-Planck equation for the probability distribution function of displacements. Can the same procedure be implemented for growth laws?

## Materials and methods
### Yeast genotypes and growth morphology
#### Snowflake yeast genotypes
Multicellular yeast groups were constructed from initially unicellular *Saccharomyces cerevisiae*. Petite yeast groups (P-) were used in all experiments except those noted below. Snowflake yeast were engineered by replacing a functional copy of *ace2* with a nonfunctional version as described in *Ratcliff et al., 2015* (these modified genotypes will be referred to as either snowflakes or Ace2KO). Under daily selection for large size through settling in liquid media, groups can arise via a single mutation in the *ace2* gene (*Ratcliff et al., 2012*; *Ratcliff et al., 2015*). When the *ace2* gene is not expressed, the final stage of cell division is not completed, and mother-daughter cells remain attached at the chitinous bud site. Since all cells are attached directly to their mothers, snowflake groups form a fractal-like branched tree collective. To measure bud scar size, we used a unicellular strain of Y55 yeast; these measurements were only used to pick parameters for snowflake yeast simulations.

#### Yeast growth morphology
*S. cerevisiae* cells reproduce by budding, a type of asexual reproduction where a new cell extrudes from the surface of the parent cell. During budding, mother and daughter cells remain attached via a rigid chitinous bond; in unicellular yeast, chitinase will degrade this bond as the last step in cell division, releasing the daughter cell and leaving behind a 'bud scar' on the mother surface and a 'birth scar' defining the proximal hemisphere on the daughter's cell surface. In all experiments, we use yeast expressing bipolar budding patterns (*Chant and Pringle, 1995*). The bipolar budding pattern is characterized by bud sites that typically do not form along the equator of the cell. Usually, the first daughter buds near the distal pole. Subsequent budding sites are typically positioned along a budding ring defined by a polar angle $\theta$ ( *Figure 1—figure supplement 1*). Some buds will 'backbud' toward the mother cell (i.e. on the proximal end of the cell), but most buds are placed on the distal side. By contrast, the azimuthal positions of all buds appears to be randomly distributed.

## Growth conditions

All experiments were performed on yeast grown for approximately 24 hr in 10 mL of yeast peptone dextrose (YPD, 10 g/L yeast extract, 20 g/L peptone, and 20 g/L dextrose) liquid medium at 30 C, and shaken at 250 rpm in a Symphony Incubating Orbital Shaker model 3,500I. All cultures were therefore in the stationary phase of growth at the time of experiments.

### Scanning electron microscopy to measure group structure

Since yeast cells have thick cell walls that limit the effectiveness of optical microscopy, we used a Zeiss Sigma VP 3View scanning electron microscope (SEM) equipped with a Gatan 3View SBF microtome installed inside a Gemini SEM column to obtain high resolution images of the internal structure of snowflake yeast groups and locate the positions of all cells. All SEM images were obtained in collaboration with the University of Illinois's Materials Research Laboratory at the Grainger College of Engineering. Snowflake yeast clusters were grown overnight in YPD media, then fixed, stained with osmium tetroxide, and embedded in resin in an eppendorf tube. A cube of resin 200 m x 200 m x 200 m (with an isotropic distribution of yeast clusters) was cut out of the resin block for imaging. The top surface of the cube was scanned by the SEM to acquire an image with resolution 50 nm per pixel (4000 × 4000 pixels). Then, a microtome shaved a 50-nm-thick layer from the top of the specimen, and the new top surface was scanned. This process was repeated until 4000 images were obtained so that the data cube had equal resolution in $x, y, z$ dimensions.

Custom image analysis scripts were written for the SEM datasets. First, a local adaptive threshold was used to binarize the image. A distance transform was used to identify the center of each cell slice in a particular 2d image. A watershed algorithm was then seeded with the cell slice centers, followed by a particle tracking algorithm to label cells across image slices. After labeling, the boundary for each cell was found, resulting in a point cloud of the exterior of each cell. Each cell was then fitted with an ellipsoid with nine fit parameters: $(x_0, y_0, z_0)$ cell center, $(a, b, c)$ cell radii, and $(\theta, \varphi, \psi)$ for cell orientation. The net rotation matrix $R$ was then found, where each column of $R$ corresponds to the direction vector of one principal axis of the ellipsoid. We consider the radii of the principal axes $(a, b, c)$ to be part of a diagonal scaling matrix $S$ which sets the ellipsoid size. Since the SEM images only capture the cell cytoplasm, each principal axis was increased in size by an additional 100 nm to account for the cell wall during visualization. Last, although there is no possible 3d 3 × 3 translation matrix, a 4 × 4 translation matrix $T$ can capture the position of the cell center $(x_0, y_0, z_0)$. Adding one additional column and row to the matrices $R$ and $S$ with the diagonal element being 1 and all other elements being 0 then means that a unit sphere centered at the origin can be mapped to any specific cell by a surface matrix $M = TRS$, and furthermore any point on the cell's surface can be mapped back to the unit sphere by the inverse of $M$. Then, the surface matrices are the only information that must be stored. From this dataset, 20 clusters of $105 \pm 51$ cells in each cluster were identified along with their intercellular mother-daughter chitin bonds. We chose to stop at $N = 20$ clusters due to the expensive nature of hand-identifying mother-daughter bonds, and since $N = 20$ clusters gave over 2000 measurements of single cells in the clusters. In following experiments with different organisms and in following simulations of different growth morphologies, we always made at least as many measurements as for these experiments of snowflake yeast.

### Petite yeast cell size and shape

We measured cellular volumes from SEM images by ellipsoid fits. The average cellular volume of petite yeast was $v_c = 17.44 \, \text{m}^3 \pm 7.33 \, \text{m}^3$. This measurement was used in our Voronoi distribution derivations. We measured the mean cellular aspect ratio to be $\alpha \equiv a/b = 1.28 \pm 0.20$.

### Bud scar size

We next measured the typical size of bud scars on the surface of Y55 yeast cells. Single cells were stained with calcafluor to highlight the chitinous bud scars (*Figure 1—figure supplement 1*). Confocal $z$-stacks were obtained on a Nikon A1R confocal microscope equipped with a 40× oil immersion objective. These images were visualized using the image processing software FIJI, and the 3d volume viewer plugin. To track the location and size of bud scars, a custom MatLab script was written to map the strongest calcafluor signals, since calcafluor makes bud scars brighter than other portions of the cell wall. Brightness isosurfaces then isolated the bud scars from the cell wall. Next, the isosurface

points were rotated to the $x - y$ plane by finding its principal components in a principal component analysis. The rotated surface points were then fit with an ellipse, returning the major and minor axes. The average of the major and minor axes returned an average interior bud scar diameter of $1.2\,\text{m}$. This value was later used in simulations of yeast groups.

### Bud scar locations

We measured bud scar positional distributions for petite yeast Ace2KO. Since the SEM does not image chitinous bud scars, we approximated bud scar positions as the closest point on a mother cell's surface to the corresponding daughter cell's proximal pole. We recorded 1990 bud scar positions in polar coordinates, as defined in *Figure 1—figure supplement 1*. There is a clearly defined polar angle for the budding ring, while the azimuthal angle is uniformly distributed. The mean and standard deviations of the two angular coordinates were $\theta = 42\% \pm 23\%$, and $\varphi = 180\% \pm 104\%$.

## Imaging *Volvox*

### Cultivation and selective plane illumination microscopy

The *V. carteri f. nagariensis* strain HK10 (UTEX 1885) was obtained from the Culture Collection of Algae at the University of Texas at Austin and cultured as previously described (*Brumley et al., 2014*). To visualise somatic cells, *V. carteri* spheroids were embedded in 1% low-melting-point agarose, suspended in liquid medium and imaged using a custom-built Selective Plane Illumination Microscope (*Haas et al., 2018*). Each somatic cell is mostly filled with a single chloroplast. Chlorophyll autofluorescence was excited at $\lambda = 561\,\text{nm}$ and detected at $\lambda = 570\,\text{nm}$. To increase the accuracy with which we identify somatic cell positions, $z$-stacks of six spheroids were acquired from three different angles (0, 120, 240 degrees) and fused as described in the following paragraph. We used data from $N = 6$ different Volvox organisms, which gave us $\sim 6000$ measurements of single-cell positions. This gave us more data than what we measured for snowflake yeast clusters.

### Registration of cell positions

Positions of cells were registered based on fluorescence intensity using custom Matlab scripts. This was achieved by carrying out a 2D convolution of each frame of the $z$-stack with a basic kernel modeling the appearance of a cell – this was set to be an asymmetric double sigmoidal function. Cell segmentation was corrected manually. $Z$-stacks taken from different angles were roughly aligned using Fiji and the Matlab function fminsearch to minimise distances between the reproductive cells. This alignment was used as starting point for alignment of the somatic cells again using fminsearch. The positions of somatic cells were merged and averaged.

## Voronoi tessellation

We used a Voronoi tessellation algorithm to measure the distribution of cell neighborhood sizes in groups. We computed both 3D and 2D Voronoi tessellations.

### 3D voronoi tessellations

First, we computed 3D Voronoi tessellations within a defined boundary. These tessellations were performed for experimental snowflake yeast data from the SEM and simulations of 3D groups using the open-source Voronoi code Voro++ (*Rycroft, 2009*), wrapped in a custom MatLab script. Voro ++ takes as input the Cartesian coordinates of the cell centers and the boundary of the shape within which to compute the tessellation. Without a boundary, all of the Voronoi cells located on the periphery would extend to infinity. We started the tessellation process by setting the input boundary to be a sphere; the Voronoi algorithm tessellated space within the spherical boundary. Then, pieces of the sphere were pared away until a Voronoi tessellation within the group's convex hull was obtained, as described in the next paragraph.

The boundary sphere was centered on the cluster's center of mass. Its radius was the distance to the farthest cell center plus an additional $5\,\text{m}$. Upon tessellation within the sphere, each Voronoi polyhedron is defined by Cartesian vertices $\mathbf{r_j}$. We group these preliminary vertices by the cells to which they correspond, so that $Q_i = \{\mathbf{r_1}, \mathbf{r_2}, ..., \mathbf{r_m}\}$ is a list of the $m$ vertices corresponding to cell $i \in [1, N]$, $N$ being the total number of cells in the organism. We next computed the cluster's convex hull, which is the smallest convex polyhedron that contains all cell centers. We then extended the vertices of

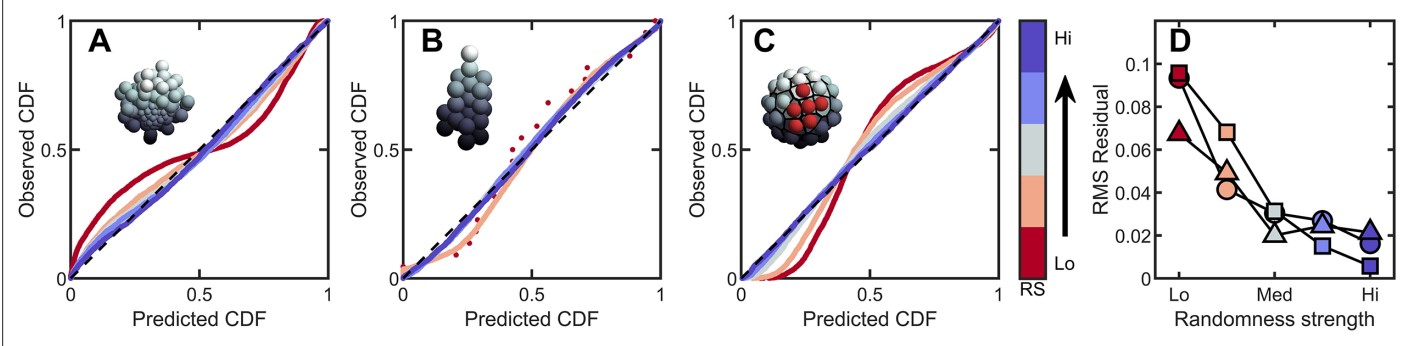

**Figure 4.** Introducing correlations and structure can break the maximum entropy distribution. In A-C are PP plots of the observed vs. predicted cumulative distribution function for three different simulations. The colors correspond to increasing levels of noisiness in the simulations, from red (strongest correlations/determinism) to blue (strongest noise). The dashed black line in each represents $y = x$, or exact predictive efficacy. (**A**) Aggregative groups with bimodal size polydispersity; noise is introduced by varying the probability that small cells reproduce into small or large cells, and vice versa. (**B**) Tree-like groups with persistent intercellular bonds that grow according to a growth plan modified by noise in cell placement. (**C**) Surface-bound groups with programmed cell death events that may be localized or randomly dispersed. (**D**) The root mean square deviation from predicted values for each simulation case. Circles are aggregative simulations from A, triangles are tree-like simulations from B, and squares are surface-bound simulations from C. Note: the discrete points in (**B**) arise because absent randomness in the cell locations the exact same cellular structure is achieved by every simulation. Therefore, the observed distribution of Voronoi volumes is noticeably discrete for this case. Upon adding randomness, the cellular structure is altered between successive simulations, and the distribution of Voronoi volumes becomes less discrete.

The online version of this article includes the following source data for figure 4:

**Source data 1.** This figure was generated from the data provided through simulations of aggregative growth with size polydispersity, tree-like growth with noise in cell position, and growth on a spherical surface with planned apoptosis.

the convex hull by $3\,\mathrm{m}$ outwards from the cluster center of mass so the boundary contained the entirety of each cell. This new boundary polyhedron, whose vertices are labeled $B$, defines the cluster boundary. We then found the intersection polyhedron, $Z_i = Q_i \cap B$ by taking the union of the dual of their vertices. This process thereby trims all Voronoi polyhedra to lie exclusively within the cluster's convex hull. The polyhedra $Z_i$ were the final Voronoi polyhedra used for the remaining data analysis.

## Voronoi tessellation on a sphere
For Voronoi tessellations of cells on the surface of simulated spheres (see *Figure 3* and *Figure 4* of the main text), we used a built-in Matlab function called 'voronoisphere' for Voronoi tessellations on a sphere.

## Voronoi tessellation on non-spherical surfaces
We also computed 2D Voronoi tessellations on surfaces embedded in 3D space using custom-written MatLab functions. This approach was used for *Volvox* experimental data. Performing this computation with *Volvox* experiments presented a challenge as *Volvox* are roughly spherical, but with varying local curvature. It was therefore necessary to compute a Voronoi tessellation on an arbitrary surface.

The first step toward generating the proper Voronoi tessellation was computing the Delaunay triangulation of the cells on the surface (the Voronoi tessellation is the dual of the Delaunay triangulation). First, we found the Cartesian coordinates of each somatic cell (as described above), and normalized these coordinates so that all cell centers laid on the unit sphere. Then, a Delaunay triangulation of the normalized points was calculated. Edges of the triangulation that cut through the unit sphere were eliminated, and edges that laid along the sphere surface were kept. This Delaunay triangulation therefore mapped out the connectivity of the somatic cells. We then projected that triangulation onto the lumpy surface. The Voronoi polygon vertices are the circumcenters of each Delaunay triangle. Further, any edge shared between two Delaunay triangles denotes an edge shared between the Voronoi vertices associated with those two triangles. We found all edges connecting the Voronoi vertices. Next, connected edges were flattened so that each Voronoi cell was a 2D polygon. This step eliminates the curvature associated with the surface of the organism. However, we found that the distribution of Voronoi areas was unaffected by taking either the planar approximation or by approximating the area by taking the local curvature into account – the average difference between Voronoi areas

when approximating the surface as a plane $A_p$ vs. approximating the surface as a spherical cap $A_s$ was found to be $\langle \frac{A_p - A_s}{A_p} \rangle = 0.001$, measured for one organism. Therefore, we used the flattened Voronoi polygons as the final tessellation shapes.

## Data analysis of voronoi measurements

In all cases, the output of the Voronoi algorithm is a list of Voronoi polytope sizes: in 3D, the measurements were the final Voronoi polyhedron volumes, while in 2D the measurements were polygon areas. Histograms of these sizes were generated to compare with the k-gamma distribution. As we observe cells in direct contact with each other, the minimum size of a Voronoi volume or area was defined by single cell measurements. For petite yeast cells, the mean cell size was calculated from the ellipsoid fits described above to be $v_c = 17.44 \text{ m}^3 \pm 7.33 \text{ m}^3$. In simulations, the minimum volume was set by the defined cell radius; in bidisperse simulations, the minimum size was set by the volume of the smallest cells.

We then calculated the expected maximum entropy distribution using only the mean and variance of the observed Voronoi volumes, $\bar{v}$ and $\sigma^2$, as inputs. Together with the minimum volume $v_c$, these measurements define $k = (\bar{v} - v_c)^2/\sigma^2$, a dimensionless shape parameter (**Aste and Di Matteo, 2008**). The maximum entropy distribution was therefore not fit to the data using, for example, a least squares method, but inferred from the first two moments of the distribution.

### Volvox

Along the surface of the *Volvox* organisms, there are gaps between some of the somatic cells due to the gonidia that lie beneath, but near the surface of the organism. These gonidia effectively occupy space on the surface, making it inaccessible to somatic cells. We excluded all Voronoi cells that intersected these gonidia gaps. We identified gaps in the soma cells by flagging Delaunay triangles with exceptionally high aspect ratios. Any Voronoi polygons that intersect the flagged Delaunay triangles were then flagged and later excluded from the dataset. The polygons were generally spatially clustered, indicating that the gonidial gaps were being correctly isolated. Roughly 90 polygons were excluded from each organism.

In *Volvox* organisms, each cell is surrounded by extracellular matrix, so cells do not contact each other. Furthermore, each of the six organisms studied varied in diameter (standard deviation in diameter was 28.2 m), yet all contained roughly the same number of somatic cells, leading to systematic differences in average surface area per cell across the organisms. Quantitatively, the coefficient of variation of the diameter of the groups was $CV_D = 0.05$, while the coefficient of variation in the number of cells in each group was roughly 10 times smaller, $CV_N = 0.006$. To counter the systematic size differences between organisms, we converted the Voronoi polygon areas into solid angles by dividing by the total surface area of each organism, $\Omega_i = A_i/S$; we then grouped all six organisms together into one histogram. We allowed the minimum solid angle, used in the k-gamma equation, to be a fit parameter in a least squares minimization procedure. There was one outlier cell with solid angle $\Omega = 0.0048$ steradians; the next two smallest cells had solid angles 0.0068 and 0.0069 steradians. We removed the outlier; the least squares minimization procedure then fit a minimum solid angle $\Omega_c = 0.0070$ steradians. We used this value for all further calculations. Just as in the 3D case, the mean and variance of the solid angle were measured to set the expected maximum entropy distribution.

## Goodness-of-fit analysis

To test how well the k-gamma distribution performed as a predictive distribution, we systematically compared its performance to three other distributions: the normal (Gaussian) distribution, the

**Table 1.** Model parameters for distribution comparisons.

| K-gamma $(\mu, k)$ | $\mu = \bar{V}$ | $k = (\bar{V} - v_c)^2/s^2$ |
|---|---|---|
| Normal $(\mu, \sigma)$ | $\mu = \bar{V}$ | $\sigma = s$ |
| Log-normal $(\mu, \sigma)$ | $\mu = \exp(\bar{V} + s^2/2)$ | $\sigma = (\exp(s^2) - 1)\exp(2\bar{V} + s^2)$ |
| Beta prime $(\alpha, \beta)$ | $\alpha = \bar{V}(\bar{V} + \bar{V}^2 + s^2)/s^2$ | $\beta = (\bar{V} + \bar{V}^2 + 2s^2)/s^2$ |

log-normal distribution, and the beta prime distribution. We used two of our datasets for this comparison: experimental values of Voronoi volumes for snowflake yeast, and simulations of snowflake yeast (which provided us with more datapoints for comparison). First, we used the measured mean $\bar{V}$ and standard deviation $s$ of the datasets to estimate each distribution's parameters. In *Table 1* are the two parameters of each distribution and how they are calculated from the measured mean and variance of the dataset.

The three distributions that we used to compare to the k-gamma distribution were chosen based upon their properties. For instance, all three comparison distributions have two parameters; also, all three chosen distributions have either semi-infinite or infinite domain. We chose the Normal distribution because it is the centrally-limiting distribution for a process with summative random errors. We chose the Log-normal distribution because it is the centrally-limiting distribution for a process with multiplicative random errors. Last, we chose the beta prime distribution arbitrarily; the goal was to make comparisons to a distribution which we have no reason to believe would accurately describe this dataset.

To systematically compare the distributions, we sampled $N$ datapoints, with replacement, from the original datasets. We varied $N$ to test the effect of finite datasets. Each time we sub-sampled the dataset, we calculated the mean and variance of the new data sample, then used those measured values to estimate the parameters of the four distributions. We then recorded the root-mean-square residual for all four cases. We plotted the root-mean-square residual for each distribution as a function of the data sample size, $N$, in *Figure 2*. We observed that the k-gamma distribution outperformed the other distributions at every sample size.

Next, we used the four distributions to predict the skewness of the simulation data. We empirically determined the first two moments of the dataset: we take these to be the first two moments for each of our distributions. Then, we used the first two moments to predict the skewness, which is related to the third moment of the distribution. We then compared this value to experimentally-measured skewness, finding that the k-gamma distribution estimated the experimentally observed skewness with only a 7% error; the other three distributions estimated the skewness with percent errors of $-100\%, 96\%,$ and $174\%$ for the normal, log-normal, and beta-prime distributions, respectively (*Figure 2*).

As an additional test, we used least-squares fitting to test the performance of the k-gamma distribution compared to the other three distributions for simulation data. In this case, we let the least-squares fitting algorithm find the best parameters to fit each distribution; then, we compared the least-squares-achieved mean and standard deviation of the fit distribution to the measured mean and standard deviation of the population. We found that the k-gamma fit reproduced the closest values to the measured mean and standard deviation (*Figure 2*).

## Cluster size distribution measurements

Cluster sizes were measured using a Beckman Coulter Multisizer 4e particle analyzer in the Cellular Analysis and Cytometry Core of the Shared User Management System located at the Georgia Institute of Technology. Petite Ace2KO clusters were taken from steady state concentration in YPD and then submerged in electrolytic fluid and passed through a $100\,$m aperture tube. The volume measured on the multisizer corresponds to the volume of electrolyte displaced by the cluster. The number of cells in each cluster was then estimated by $N = V/v_c$, where $V$ is the volume of organism measured by the Coulter Counter, and $v_c$ is the average cell volume from SEM measurements, $v_c = 17.44\,$m$^3$.

## Cumulative distribution function statistics

To quantify goodness-of-fit for predicted maximum entropy distributions, we compared the predicated cumulative distribution function (CDF), $F(x)$, to the empirical CDF, $F_i$, using P-P plots. Exactly predicted points will lie on the line $y = x$ in these plots. We measured the root-mean-square residual from the line

$$r_{RMS} = \sqrt{\langle (F_i - F(x))^2 \rangle},$$ (12)

## Measurements of $\Psi_6$ in *V. carteri*

From the light sheet images of *Volvox*, we obtained the Cartesian coordinates of each somatic cell. From Delaunay triangulation, we then obtained a list of every cell's closest neighbors. Each cell and its *NN* nearest neighbors did not generally lie in a plane due to local curvature of the *Volvox* surface. We therefore calculated in-plane and out-of-plane components using principal component analysis. The in-plane components were then used to write the positions of each nearest neighbor in polar coordinates. The formula for calculating $\Psi_6$ is

$$\Psi_6 = |\langle \frac{1}{NN} \sum_{j=1}^{NN} e^{6i\theta_j} \rangle| \tag{13}$$

where $\theta_j$ defines the polar angle coordinates around the cell of interest and $\langle ... \rangle$ denotes averaging over all cells. We calculated $\Psi_6$ separately for each of six different organisms; we report $\Psi_6 = 0.03 \pm 0.01$.

### Correlation of voronoi areas

In *Volvox* organisms, we calculated the spatial correlation of polygon areas. First, we extracted the list of cell neighbors from the Delaunay triangulation of the organism surface. Nearest neighbors were designated as living a network distance of 1 away from a cell of interest; next nearest neighbors live a network distance of 2 away from the cell of interest, etc. The number of neighbors a network distance of $Q$ away is then $J(Q)$, which is empirically determined. The network correlation function is then

$$C(Q) = \frac{\langle (\Omega - \langle \Omega \rangle) Y_Q \rangle}{\sigma_\Omega \sigma_{Y_Q}} \tag{14}$$

where $Y_Q = J(Q)^{-1} \sum_j (\Omega_j - \langle \Omega \rangle)$ is the average deviation of the solid angle of a given polygon's neighbors from the mean. The standard deviation of the solid angle across the population is $\sigma_\Omega$, and $\sigma_{Y_Q}$ is the standard deviation of $Y_Q$ across the population.

## Simulation methods

### Simulations of snowflake yeast groups

Simulations of snowflake yeast groups were adapted from previously published work by *Jacobeen et al., 2018a*; *Jacobeen et al., 2018b* that found simulations of snowflake yeast growth morphology accurately replicated experimentally measured cellular packing fractions and average group sizes. In the present work, cells were modeled as prolate ellipsoids of revolution with a semi-major axis $a = 2.88\,\text{m}$ and semi-minor axis $b = 2.29\,\text{m}$, characterized by the aspect ratio $\alpha \equiv a/b = 1.26$. Each generation, every cell attempted to reproduce; however, if new cells closely overlapped with existing cells (i.e. their bud scars are closer than $1.2\,\text{m}$), they were eliminated. Setting the number of generations (for example, 7) sets the maximum possible number of cells in the group at the end of the simulation ($2^7 = 128$), and roughly sets the expected number of cells in the group ($\sim 100$). In our simulations, cells were 80% likely to bud first from the distal pole (i.e. $\theta = 0 \pm 10$ degrees). Subsequent cells budded at a polar angle $\theta$, and with an azimuthal angle randomly chosen from a uniform distribution $\varphi \in [0, 2\pi]$; in other words, after the first bud, cells generally appeared along a 'budding ring'. There was a 20% chance that the first bud would appear along this budding ring instead of exactly at the pole. After 3 bud scars, there was a 50% chance that new cells bud on the proximal side ($\pi - \theta$) instead of the distal side. The orientation of the new cell is determined by the surface normal to the mother cell at the position of the bud site; the major axis of the new cell lies along the surface normal.

To compare exhaustively the distribution of Voronoi volumes between simulations and the k-gamma distribution, we simulated $9,100$ clusters. In each simulation, clusters were allowed to grow for 7 generations of cell division, corresponding to an average of $94.2 \pm 10.9$ cells per cluster. The budding ring was defined by the polar angle $\theta = 45\%$, a close approximation to the experimentally measured mean polar angle. These simulations did not include intercellular forces. The cell centers were recorded and then Voronoi tessellations were made within each cluster's convex hull.

### Simulations of *V. carteri*

We simulated *10 Volvox*-like groups with $N = 1000$ cells confined to the surface of a sphere. Cells were placed on the surface of a sphere of unit radius by randomly selecting polar and azimuthal coordinates

in a Poisson point process. The process proceeds as follows: each new cell was randomly placed, and its distance from all other cells was calculated. If the new cell is within a threshold distance $d$ from any existing cell, it was removed and a new cell was placed elsewhere on the spherical surface. This process was iterated until all 1000 cells were placed. We chose a minimum separation distance of $d = 0.088$, which allowed reasonably rapid convergence. We then calculated the Voronoi tessellation and the correlation function as described above. The $N = 10$ groups, each with 1000 cells, returned 10000 single cell measurements.

## Simulations of two additional growth morphologies

We next sought to model two additional classes of growth morphologies: sticky aggregates and cells contained within a maternal membrane. In both simulations, cells were modeled as spheres with unit radius.

## Aggregative groups

First, we considered a multicellular model of sticky aggregates, mimicking group formation in, for example, flocculating yeast and bacterial aggregates. In our simulations, groups were grown from a single cell. New spherical daughters appeared at a polar angle $\theta$ and azimuthal angle $\varphi$. Within each step, there was stochasticity in the budding location: cells would appear at $\theta = \theta_0 \pm 15\%$. The azimuthal angle was always drawn from a uniform distribution on the interval $[0\%, 360\%]$. We simulated $N = 50$ aggregative groups, each with 64 cells, for a total of $gt_{3000}$ measurements of single-cell volume.

Cells interacted with both steric and attractive interactions in overdamped dynamics. Steric interactions were modeled through a harmonic potential when two cells overlapped, with a cutoff once cells were no longer overlapping. That is, for two cells and $j$ (radii $R_i$ and $R_j$) separated by the vector $\mathbf{r}_{ij} = \mathbf{r}_j - \mathbf{r}_i$, the steric force acting on cell from cell $j$ is

$$\mathbf{F}_{ij} = \begin{cases} 0 & |\mathbf{r}_{ij}| > (R_i + R_j) \\ \kappa_s \left( |\mathbf{r}_{ij}| - (R_i + R_j) \right) \hat{\mathbf{r}}_{ij} & |\mathbf{r}_{ij}| \leq (R_i + R_j) \end{cases} \tag{15}$$

Attractive interactions (i.e. sticky, aggregative bonds) were also modeled through a harmonic potential, but these interactions had both a lower bound and upper bound cutoff.

$$\mathbf{G}_{ij} = \begin{cases} 0 & |\mathbf{r}_{ij}| > 2(R_i + R_j) \\ -\kappa_a(|\mathbf{r}_{ij}| - a(R_i + R_j))\hat{\mathbf{r}}_{ij} & (R_i + R_j) \leq |\mathbf{r}_{ij}| \leq 2(R_i + R_j) \\ 0 & |\mathbf{r}_{ij}| < (R_i + R_j), \end{cases} \tag{16}$$

where $a$ sets the location of the attractive well minimum. We used $a = 0.9$, so that the attractive interactions allow a small amount of cell overlap.

## Size polydispersity

In simulations in which we introduced size polydispersity, cells were allowed to reproduce into two separate sizes, $R_1 = 1$ and $R_2 = 2$. The probability of budding cells of the same size as the mother cell is denoted $\xi$. When $\xi = 1$, the mother cell always produces cells of the same size, while when $\xi = 0.5$, there is a 50% chance that the mother cell produces a cell of size $R_1$ or $R_2$, independent of the radius of the mother. Simulations were seeded with a pair of contacting cells, one each of the two radii. The simulation then proceeded with subsequent rounds of cell division and mechanical relaxation. We varied the polydispersity parameter $\xi$ in even increments from $\xi = 0.5$ to $\xi = 1.0$ for a total of 6 different simulations. In each simulation, we simulated $N = 50$ organisms, each with 64 cells, for a total of 3200 cells per simulation, and almost 20,000 cells total.

## Groups confined within a membrane

In another common mode of group formation, cells divide repeatedly within a confining membrane. This type of group formation has been observed in experimentally-evolved multicellular algae derived from unicellular *Chlamydomonas reinhardtii* (*Herron et al., 2019*), and is reminiscent of both baeocyte

production in *Stanieria* bacteria (*Angert, 2005*), and neoproterozoic embryo fossils (*Xiao et al., 1998*). In a simulation model, we adopted the essential components of this class of growth: groups grow from a single spherical cell, cells divide stochastically, and cells interact sterically with both a maternal cell wall and each other. Typically, palintomic cell division occurs rapidly, meaning that the packing fraction remains the same within the maternal cell wall. We simulated this by increasing the radius of the cell membrane after each cell division, but before allowing any mechanical relaxations.

Steric forces between a cell and the maternal cell wall were modeled as being proportional to the non-overlapping volume of the cell and the maternal cell wall. In other words, if a cell is not contacting the membrane, there is no force acting on it. However, if the cell is contacting the membrane, the force is proportional to how much of the cell volume lies outside the membrane. Each cell was assigned volume $v_c = 4/3 * pi$. The overlapping volume of the cell and the membrane is labeled $v_i$. The force the cell experiences from the membrane is then

$$\mathbf{F}_i = \kappa_m (v_c - v_i)\hat{\mathbf{r}}_i, \tag{17}$$

where $\hat{\mathbf{r}}_i$ is a unit vector pointing to the center of the maternal membrane. Additionally, steric interactions between cells were calculated as described above for aggregative groups.

We simulated $N = 200$ groups growing in a confining membrane, each with 64 cells in the group, for a total of 12800 measurements of single cells.

## Groups confined to a spherical surface

Some groups form by arranging cells around a central core of extracellular matrix (ECM). To simulate such groups, we modeled a sphere of ECM with cells arranged randomly along the surface. Cell positions were chosen by selecting a position in spherical coordinates from uniform polar $\theta \in [0, \pi]$ and uniform azimuthal $\varphi \in [0, 2\pi]$ distributions. The only rule implemented in cell placement is that no two cells can be located closer than two cell radii from one another. If a new cell is chosen to be located too close to any existing cells, it is eliminated and a new position is chosen. We iterated this process until $N$ cells were placed on the ECM surface.

First, we chose to place $N = 50$ cells on the surface. Therefore, the maximum cell radius allowing all 50 cells to be placed is 0.283 units (where the total sphere has unit radius). We chose the cell radius to be 0.1980 units, which allowed for reasonably rapid random placement of all 50 cells (other choices of cell radii demonstrate qualitatively similar results).

## Simulated cellular apoptosis

In simulations with apoptosis events, cell death occurred after group generation. Briefly, groups were generated by iterated generations of cell division starting from a single cell. After this process, one cell was chosen at random to die. Then, all cells within a localization radius $R$ were flagged. Of the flagged cells, 9 more were chosen at random to die. Therefore, small localization radii correspond to highly localized death events, where 10 juxtaposed cells may die together. As the localization radius increases, there are more flagged cells, and therefore more randomness in cell death. All other cells were unaffected by the cell death process. We simulated six different cell death radii. Each cell death radius simulation included 100 organisms simulated, each with 43 cells after the cell death process. Therefore, each cell death radius $R$ includes 4300 single cell measurements, for a total of 25,800 measurements.

## Tree-like groups with precisely defined cell placement/location

We also investigated groups with precisely defined growth patterns. The spherical cells were held together with fixed, chitin-like bonds. The first cell was placed at the origin. It then proceeded to bud 3 daughter cells, each of which also budded subsequent cells. The exact budding pattern is described below.

Daughter cells were placed as follows. In spherical coordinates on the surface of the mother cell, the first daughter cell was placed at ($\theta = 0 \pm \eta$, $\phi = 0 \pm \eta$), the second at ($\theta = 90 \pm \eta$, $\phi = 90 \pm \eta$), and the third at ($\theta = 90 \pm \eta$, $\phi = 270 \pm \eta$), where $\eta$ is the strength of random noise added.

The first daughter cell's coordinate system was rotated $90\% \pm \eta$ around the $z$-axis from the mother cell; in other words, for the first daughter cell, $x \to x'$, $y \to y'$, and $z \to z'$, where $\mathbf{x}' = \mathbf{R}_z(\pi/2 + \eta)\mathbf{x}$, $\mathbf{y}' = \mathbf{R}_z(\pi/2 + \eta)\mathbf{y}$, and $\mathbf{z}' = \mathbf{R}_z(\pi/2 + \eta)\mathbf{z}$, and $\mathbf{R}_z$ is the rotation matrix around the $z$-axis. This daughter

cell then proceeded to bud daughters in the exact same pattern as its mother; however, because its local coordinates were rotated, the budding positions were also rotated 90% with respect to the mother cell's buds. This process was iterated for five generations of cell division. When $\eta = 0$, this corresponds to only three cells overlapping three other cells. The three overlapping cells were then removed.

After each round of cell division, cells were allowed to relax mechanically in overdamped dynamics according to steric repulsive interactions and sticky, rigid bond interactions to their mother cell. The steric interactions were the same as described above. Fixed bond interactions were modeled as follows. When new cells appear, they incur a bud scar on the mother cell's surface and a birth scar on the daughter cell's surface. The positions $\mathbf{r}_{bu}$ and $\mathbf{r}_{bi}$ of the bud scar and the birth scar were recorded and tracked. The vector pointing from the bud scar (on the mother's surface) to the birth scar (on the daughter's surface) is $\mathbf{r} = \mathbf{r}_{bi} - \mathbf{r}_{bu}$. Then, the force acting on a cell from it's mother cell is

$$\mathbf{F}_{mother} = \kappa \left( |\mathbf{r}| - 2 \right) \hat{\mathbf{r}} \tag{18}$$

where $\kappa$ was the chitin bond strength. In addition, cells experienced forces from all of their daughter buds (given by the same relationship and the same chitin bond strength). The initially seeded cell did not experience forces from a mother cell.

For $\eta = 0$ (i.e. no noise), the distribution of Voronoi volumes was visually discontinuous, since cells could only access a finite number of local configurations. As the noise strength increased, the maximum entropy predictions were gradually recovered. We simulated 11 different noise strenths, each with 150 organisms, each with 28 different cells simulated, for a total of 46,800 measurements.

## Acknowledgements

Core Facilities at the Carl R Woese Institute for Genomic Biology. WCR was supported by NIH grant 1R35GM138030. This work was funded in whole, or in part, by the Wellcome Trust (Grant 207510/Z/17/Z; REG & SSH). For the purpose of open access, the authors have applied a CC BY public copyright license to any Author Accepted Manuscript version arising from this submission. This work was also supported in part by Established Career Fellowship EP/M017982/1 from the Engineering and Physical Sciences Research Council (REG).

## Additional information

### Competing interests

Raymond E Goldstein: Reviewing editor, eLife. The other authors declare that no competing interests exist.

### Funding

| Funder | Grant reference number | Author |
| --- | --- | --- |
| National Institutes of Health | 1R35GM138030 | Will Ratcliff |
| Wellcome Trust | 207510/Z/17/Z | Stephanie S Hohn<br>Raymond E Goldstein |
| Engineering and Physical Sciences Research Council | EP/M017982/1 | Raymond E Goldstein |
| National Institutes of Health | 1R35GM138354-01 | Peter J Yunker |
| Engineering and Physical Sciences Research Council | Vacation Bursary | Hannah R Sleath |
| John Templeton Foundation | A009723003 | Raymond Goldstein<br>Stephanie S Höhn |

| Funder | Grant reference number | Author |
|--------|------------------------|--------|

The funders had no role in study design, data collection and interpretation, or the decision to submit the work for publication.

## Author contributions

Thomas C Day, Conceptualization, Data curation, Formal analysis, Investigation, Software, Visualization, Writing – original draft, Writing – review and editing, Methodology; Stephanie S Höhn, Conceptualization, Funding acquisition, Investigation, Methodology, Project administration, Software, Supervision, Validation, Visualization, Writing – original draft, Writing – review and editing; Seyed A Zamani-Dahaj, Investigation, Methodology; David Yanni, Formal analysis, Investigation, Validation, Writing – review and editing; Anthony Burnetti, Data curation, Investigation, Writing – review and editing; Jennifer Pentz, Data curation, Formal analysis, Investigation, Software, Writing – review and editing; Aurelia R Honerkamp-Smith, Conceptualization, Formal analysis, Investigation, Methodology, Software; Hugo Wioland, Data curation, Investigation, Methodology, Software, Visualization; Hannah R Sleath, Investigation, Software; William C Ratcliff, Peter J Yunker, Conceptualization, Formal analysis, Funding acquisition, Resources, Supervision, Writing – original draft, Writing – review and editing; Raymond E Goldstein, Conceptualization, Formal analysis, Funding acquisition, Investigation, Methodology, Project administration, Validation, Writing – original draft, Writing – review and editing

## Author ORCIDs

Thomas C Day ⬡ http://orcid.org/0000-0003-4681-9348
Stephanie S Höhn ⬡ http://orcid.org/0000-0003-1815-705X
Raymond E Goldstein ⬡ http://orcid.org/0000-0003-2645-0598
Peter J Yunker ⬡ http://orcid.org/0000-0001-8471-4171

## Decision letter and Author response

Decision letter https://doi.org/10.7554/eLife.72707.sa1
Author response https://doi.org/10.7554/eLife.72707.sa2

# Additional files

## Supplementary files

- Transparent reporting form
- Source code 1. Matlab code for simulations of aggregative groups.
- Source code 2. Matlab code for simulations of groups grown confined within a membrane.
- Source code 3. Matlab code for simulating groups confined to a spherical surface.
- Source code 4. Matlab code for simulations of snowflake yeast, where cells bud at various angles and remain attached permanently to their mother cell.
- Source code 5. Matlab code for simulations of prescribed tree-like growth. The cells grow at exact positions/angles with noise strength determined by input parameters.

## Data availability

Figures 1 & 3 source data: Experimental data files enumerating the cell centers positions for each organism sampled. The folders are subdivided into those on snowflake yeast (from SEM studies) and Volvox (from lightsheet studies). Each subdirectory contains an explanatory README file. Figures 2, 4 & 5 source data: Simulation data (enumerating the cell center positions) for the six classes of numerical studies; aggregation, apoptosis, polydispersity, snowflake yeast growth, tree-like growth, and Volvox growth. Each subdirectory contains an explanatory README file.

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

## Appendix 1

It may appear surprising that the distribution of cell volumes is not governed by the Central Limit Theorem (CLT), i.e. the volumes are not distributed normally. After all, Voronoi polytope volumes are generated from many randomly interacting pieces - should not these many different random fluctuations sum to a CLT-like scenario? A simple comparison between the modified gamma distribution, a normal distribution, and a log-normal distribution shows in fact that both the normal distribution and the log-normal distribution fail to capture essential characteristics of the volume packing, while the k-gamma distribution does. For snowflake yeast, the reason for this disagreement is that as each new cell is added to a cluster, it changes the entire volume distribution, since the new cell occupies space which was previously unoccupied. It therefore changes the volumes of all its nearest neighbors; if they flex to accommodate the new cell, then those neighbors change the Voronoi volumes of their neighbors, and so on. Therefore, adding a new cell does not sample the same distribution as before - the distribution itself changes, rendering the limit inapplicable.

In the case of the *Volvox*, the somatic cells are originally connected together only by cytoplasmic bridges, forming a small sphere. As the ECM is generated the sphere "inflates". This process, in which many random fluctuations in the amount of ECM excreted by each cell over time can integrate together, seems appropriate for CLT-like arguments. However, it is worth noting that the cells are generally locally oriented with a hexagonal symmetry. In order to maintain a non-wrinkled surface, more ECM must be secreted in some local regions, such as the corners of the hexagons, than in other places, such as at the hexagon edges. Since there is no local wrinkling observed, the secretion of ECM from the somatic cells cannot be a completely random process orientationally. In other words, the ECM excretion process is controlled, which implies that the CLT does not properly capture the sampling space. Instead, the cells inevitably occupy positions on the surface of the sphere that vary from organism to organism; the maximum entropy distribution of their Voronoi areas is then the k-gamma distribution.

