## [Editor Report]

This work uncovers a simple but far-reaching statistical principle that describes the geometry of cell packing in snowflake yeast and green algae. It draws on ideas from granular physics to offer new insight into universal rules of multicellular geometry that are otherwise easily obscured by the cell-scale idiosyncrasies of the different biological systems.

---

## [Decision Letter]

**Decision letter after peer review:**

Thank you for submitting your article "Cellular organization in lab-evolved and extant multicellular species obeys a maximum entropy law" for consideration by *eLife*. Your article has been reviewed by 2 peer reviewers, and the evaluation has been overseen by a Reviewing Editor and Aleksandra Walczak as the Senior Editor. The reviewers have opted to remain anonymous.

Essential revisions:

Both reviewers praise the quality of the paper and the relevance of the results. There are not, in their opinion, critical aspects in the manuscript to be further addressed. However, they suggest a number of revisions, which would improve the clarity and presentation of the work. In particular, both reviewers think that more discussion is needed of the maximum entropy model, e.g. whether additional information on higher order structures or morphology related correlations might lead to more effective statistical models. Referee 2 also advises for more contextualization of the results, and a wider discussion about their generality.

Below the full reports, the authors are kindly invited to take into account the referees' comments in a revised version.

*Reviewer #1:*

The manuscript by Day and colleagues investigates the geometry of cell packing in two multicellular eukaryotes (snowflake yeast and green algae). Using a combination of experiments and models drawn from statistical physics, they show that the distribution of cellular neighborhood volumes follows a simple universal form – a modified gamma function – that arises from a maximum entropy argument. Using simulations of different growth processes, they then show that these universal distributions are ubiquitous-arising, for example, even in correlated systems as long as there is a minimal level of noise. Finally, they show how these principles contribute to emergent evolutionary features (specifically group size distributions) in snowflake yeast, and use simple theoretical models to argue that fluctuations, while inherently stochastic, give rise to robust structures that do not depend sensitively on the microscopic and biological features of the system.

This paper is a beautiful example of how simple biophysical models can provide fundamental and unifying insight into complex biological systems. It is well written, addresses an important and timely topic, and raises intriguing questions about the balance between "regulated" biology and simple statistical physics as selective forces for evolution.

I have several comments for the authors to consider, at their discretion. Overall, I really enjoyed this paper and learned a great deal from it.

– The manuscript offers an interesting guiding principle that describes two considerably different biological systems. As the authors show in simulations, the principle is expected to hold over a broad range of conditions, but of course not universally (though even small levels of stochasticity broaden the range of applicability). I think the paper could be improved by expanding on the discussion of these limitations. In particular, it is not clear to me exactly how surprising it is to see "good" fits to a 2-parameter distribution of this sort (or more generally, what level of "good" we should expect of the fits in finite data sets like these). The authors address this issue in part by showing fits to other distributions, which is nice. But I wonder if it would be helpful to also include (or at least discuss) more systematic model selection. To be clear, I find the analysis quite convincing as is. But I am trying to get my head around the limitations, and in particular, to get a feel for how likely one is to see similar "goodness of fit" results using other distributions with a relatively small number of parameters.

– Related to the previous point: one approach might be to construct a type of "null" model from the data, perhaps by systematically shuffling the data in some way and then bootstrapping to evaluate the likelihood of achieving fits of similar quality.

– Have the authors considered trying to systematically quantify the impact of including higher-order structures in the max ent model? For example, one could perhaps use multi-information metrics (https://journals.aps.org/prl/abstract/10.1103/PhysRevLett.91.238701) to evaluate the extent to which higher-order features of the data are relevant / necessary; the idea is essentially to construct maximum entropy models with various levels of data complexity "built in" and then evaluate (perhaps with an info-theoretic style metric) the extent to which that complexity improved the model. Perhaps something similar has been done for granular materials to capture higher-order statistics of packing? I ask this primarily out of curiosity, not as a serious criticism of the current approach. A discussion of this point might add to the paper.

*Reviewer #2:*

Day, Thomas C. et al. investigate the geometrical statistic of cell packing in multicellular organisms. Using a maximum entropy prediction originally developed in the study of granular materials, the authors show that the statistics of cell packing imposes a robust physical, entropic constraint on the geometrical arrangement of cells. Strikingly, the authors show that both snowflake yeast evolved under lab conditions and wild-type Volvox, which develop according to very different processes and have disparate overall morphology, both exhibit cell packing statistics consistent with the maximum entropy predictions. They then use simulations to show that entropic cellular packing can arise from various modes of multicellular development due to randomness in cell positions and that substantial deviations from entropic packing arise only in the case of low developmental noise (randomness) and strong correlations in cell positions. Finally, the authors use theory and measurements from experiments with yeast to show how maximum entropy statistics dictate size heritability in simple multicellular systems. Together, their results support the perhaps counterintuitive result that developmental randomness can actually underpin developmental reproducibility, in this case reproducibility in the geometry of cell packing in terms of the free space associated with individual cells within a multicellular structure. This work contributes the identification and new consideration of a fundamental physical constraint of particular relevance to the evolutionary origins of multicellularity and to multicellular morphogenesis in general.

The conclusions of this paper are well supported by the rigorous analysis of data and simulations.

Work on the evolution of multicellularity has traditionally focused on molecular and genetic mechanisms, but because multicellular morphogenesis is an inherently physical process, biophysical studies provide an important complementary perspective. A particular strength of this paper is that insights are derived from theory that requires few, but specific, conditions be met in order to be satisfied, and therefore stands to apply generally to diverse multicellular systems, irrespective of many differences between them. The combination of empirical results from disparate multicellular systems in conjunction with simulations encompassing an expanded set of multicellular morphologies and growth processes compellingly support the generality of the insights. Beyond simply speculating about the implications of entropic packing on the function of multicellular systems, the authors demonstrate impact or lack thereof on aspects of form and function in multicellular yeast and Volvox. Importantly, simulations allowed the authors to investigate in detail the robustness of theoretical predictions in terms of deviations from theory arising from developmental processes. In addition to providing new insight, this work lays the foundation for the exciting possibility of inferring aspects of developmental dynamics and regulation simply by observing the statistics of cell packing in an organism, which could be of great use in comparative evo-devo studies where developmental processes are difficult or impossible to observe.

While the work is very strong overall, there are a few caveats to consider, primarily concerning the simulations. Multicellularity takes many forms by many different processes among eukaryotes. While simulations do cover a range of different morphologies and developmental processes found in nature, factors not explicitly addressed such as constraint or patterning by secreted extracellular matrix, differences in cell shape, cell migration, and others can lead to different kinds of multicellular form. The extent to which potential correlations imposed by diverse morphologies might lead to deviations from theoretical maximum entropy predictions, and how robust those deviations might be to noise is not entirely clear. Additionally, the randomness strength in simulations from high to low, while reasonable, does not appear to be grounded in empirical characterization of randomness strength in developmental processes across biological systems. Ultimately, although they leave some uncertainty as to the generality of the results, these limitations do not contradict or significantly diminish the key claims of the paper.

Comments for the authors:

1) The simulation results are compelling but left me with some questions. To what extent do the morphologies and processes investigated by simulations address the diverse forms of multicellularity encountered across eukaryotes? To what extent does the overall shape of the multicellular structure affect the cell packing distribution (e.g. multi-lobed structure as in Zoothamnium niveum, dichotomous branching as in Dinobryon, something with an undulating boundary)? Are there any examples of simple multicellular eukaryotes that might exhibit very strongly correlated cell positions? What is known about randomness strength or precision in developmental processes in biological systems, and if anything is known, how does this compare to values in simulations? Providing a bit more contextualization or motivation for specific choices in simulations could help address these questions and would support the generality of conclusions drawn from the simulations. Although I am convinced that the results hold for a broad range of multicellular architectures and do not think that the possible existence of a few edge cases contradicts the main conclusions of the work, it is not entirely clear to me that the effects of growth morphology, connection topology, and dimensionality have been accounted for.

2) The sections titled "Multicellular motility is robust to cellular area heterogeneity" starting on p. 11 is slightly perplexing. It is certainly interested, and I see that it addresses a question that may arise from analysis of Volvox cell packing, but in its current form, I do not believe it contributes substantially to the key points of the paper. The introduction section seemed to imply that the results would demonstrate that fluctuations in cell packing may play a role in the evolution of multicellular systems, but as I understood them, the results suggest that fluctuations do not affect motility, at least implying that there should be little to no effect on any aspect of fitness related to motility. It is possible that there could be other aspects of organismal fitness related to cell packing, so while these results are consistent with cell packing fluctuations not necessarily impeding or constraining the evolution of multicellularity, they do not strongly support that conclusion. Perhaps contextualizing the results a bit more in terms of key points of the paper while reporting them and referring to them in the Discussion section might help the reader better appreciate their significance within the context of the paper overall.

3) I might suggest removing or otherwise modifying the phrase "highly-evolved" (p.14) as its meaning is unclear, has connotations of evolutionary teleology, and clashes with the fact that all extant organisms share an evolutionary history of equal length. Maybe something such as "organisms with highly-regulated development" may be more appropriate.

4) Is anything known about the source of correlated subregions of cells in Volvox? Do the authors have any ideas about this? Either way, it would be interesting to know and may warrant at least a small comment in the text.

5) In the author list, SSH is missing an asterisk to denote corresponding authorship.

6) An "e" is missing in "surface" in the caption for Figure 2B.

7) I believe the dotted red line in Figure 4B should be a solid line to match those in panels A and C.

---

## [Author Response]

Reviewer #1:[…] I have several comments for the authors to consider, at their discretion. Overall, I really enjoyed this paper and learned a great deal from it.– The manuscript offers an interesting guiding principle that describes two considerably different biological systems. As the authors show in simulations, the principle is expected to hold over a broad range of conditions, but of course not universally (though even small levels of stochasticity broaden the range of applicability). I think the paper could be improved by expanding on the discussion of these limitations. In particular, it is not clear to me exactly how surprising it is to see "good" fits to a 2-parameter distribution of this sort (or more generally, what level of "good" we should expect of the fits in finite data sets like these). The authors address this issue in part by showing fits to other distributions, which is nice. But I wonder if it would be helpful to also include (or at least discuss) more systematic model selection. To be clear, I find the analysis quite convincing as is. But I am trying to get my head around the limitations, and in particular, to get a feel for how likely one is to see similar "goodness of fit" results using other distributions with a relatively small number of parameters.

We completely agree that assessing goodness of fit is crucial, despite the fact that doing so is difficult for complex, non-linear, non-monotonic functions. Thanks to these comments, we have clarified our discussion of the topic and added a new analysis (described in detail in response to the next question). Thank you for this suggestion; we believe addressing it has strengthened the manuscript.

We agree that it is, in general, unclear how well one should expect 2-parameter distributions to perform and what exactly distinguishes a good 2-parameter distribution from a bad 2-parameter distribution for our datasets. To rectify this, we consider four different 2-parameter distributions that use the empirically measured mean and standard deviation (as well as the measured minimum cell size); they are the k-gamma, normal, log-normal, and beta-prime distributions (in the original text, we compared the data only to the normal and log-normal distributions). We do not calculate least-squares fits for these distributions, but simply use the empirically measured values of the mean and variance to extract the two relevant parameters of each distribution. For both experiments and simulations, the k-gamma distribution gives the best match to our data (Figure 2 Supplement 1). Further, as our simulations are less data-limited, we can confirm that the k-gamma distribution is significantly more accurate than the other distributions at predicting the frequency of large Voronoi volumes.

These four distributions were purposefully chosen. The rationale for the k-gamma distribution with regards to maximum entropy packing is detailed in the text. The normal, log-normal and beta-prime choices are discussed below.

We chose the normal distribution as it is the maximum entropy distribution given only the mean and variance of a population; furthermore, it sits as the limiting distribution according to the central limit theorem. Therefore, should random fluctuations add together completely independently, we might expect to observe a (truncated) normal distribution of the volumes. The absence of agreement between the data and this distribution implies that there is an additional feature that must be considered; namely, that cell volumes are not completely independent of one another but rather must sum to match a total volume (this is, of course, a requirement for the k-gamma maximum entropy distribution).

Similarly, the log-normal distribution was chosen for comparison due to its MaxEnt and Central-Limitlike properties. In essence, the log-normal distribution is the central limit theorem result for the logarithm of a variable. Further, volumes must be positive numbers, and the log-normal distribution only exists in the positive domain. Log-normal distributions are observed in many natural systems such as ecology, physiology, geology and more, and have recently been shown to describe organism swimming speeds quite well [6]. They define a good null model for any complex process with many interacting elements, particularly when there is multiplicative noise. By observing that the k-gamma distribution outperforms the log-normal distribution, we recognize that the volumes are not multiplicatively independent of one another.

Last, we chose to compare the performance of the k-gamma distribution to an arbitrary distribution with two parameters: we chose the beta-prime distribution, which has two parameters (*α* and *β*) that can be computed from the mean and variance of the population. It also has a semi-infinite domain. As we might expect, the arbitrarily chosen beta-prime distribution underperforms compared to the k-gamma distribution.

Therefore, showing these four distributions together allows us to compare four “minimal” models, and to demonstrate that the k-gamma distribution is the most accurate match for our experiments and simulations.

– Related to the previous point: one approach might be to construct a type of "null" model from the data, perhaps by systematically shuffling the data in some way and then bootstrapping to evaluate the likelihood of achieving fits of similar quality.

Thank you for this suggestion. We agree that this is an excellent approach to justify systematically the chosen distributions. For four 2-parameter distributions (k-gamma, normal, log-normal, and beta-prime), we applied a bootstrapping/resampling approach to understand how much error one might expect to see in these fits given the finite data sets. We applied this approach to experimental and simulated snowflake yeast Voronoi volumes. We took variable number subsamples of these datasets and measured how these four 2-parameter distributions perform. For each sample size, we computed the dataset’s mean and standard deviation, then used these parameters to calculate the necessary parameters for each of the distributions (i.e., we again did not do any “fitting” in a least-squares sense). By calculating the root-mean-square residual of the cumulative distribution functions, we found that the k-gamma distribution always outperforms the others (see Figure 2 Supp 1c). Further, the k-gamma distribution error continually decreases with increasing sample size (albeit slowly for large sample sizes), while the error plateaus at a larger value for all other distributions.

Next, we used the four distributions to predict the skewness of the simulation data. We empirically determined the first two moments of the dataset: we take these to be the first two moments for each of our distributions. Then, we used the first two moments to predict the skewness, which is related to the third moment of the distribution. We then compared this value to experimentally-measured skewness, finding that the k-gamma distribution estimated the experimentally-observed skewness with an error of only 7 percent, while the other three distributions (normal, log-normal, and beta-prime) estimated the skewness with percent errors of −100%, 96%, and 174%, respectively.

As an alternative approach, we test the ability of our selected distributions to perform in a least-squares fit. In other words, we fit the k-gamma, log-normal, Gaussian, and beta-prime distributions to the observed dataset. Then, we compare the values of the fitted parameters (namely, the mean and variance) to the observed values. Distributions that more closely fit the observed data are expected to produce closer estimates of the observed moments. Indeed, upon least-squares fitting the simulated dataset, we observe that the k-gamma distribution fit to the data most closely matches both the observed mean and the observed standard deviation, indicating that it is not only the distribution that best describes the tail of the distribution (as visualized in Supplementary Figure 2) but also the best fit for the distribution when considering the first two moments.

Figure additions: We revised and appended two supplemental figures for this response and the previous one. We included one figure on the probability distribution and cumulative distribution functions for all datapoints (Figure 2—figure supplement 1a,b), and then an additional panel on the bootstrapping analysis (Figure 2—figure supplement 1c). Then, we added a figure detailing the comparison of the estimated skewness from each distribution to the true measured values, and then of the measured mean and variance to the least-squares fit mean and variance for all four distributions (Figure 2—figure supplement 2).

Text additions: We added a subsection to the Methods section titled “Goodness-of-fit analysis” in which we present these results. In this section, we describe our reasons for choosing the three comparison distributions, indicate the parameters involved in each, and finally compare the performance of each.

– Have the authors considered trying to systematically quantify the impact of including higher-order structures in the max ent model? For example, one could perhaps use multi-information metrics (https://journals.aps.org/prl/abstract/10.1103/PhysRevLett.91.238701) to evaluate the extent to which higher-order features of the data are relevant / necessary; the idea is essentially to construct maximum entropy models with various levels of data complexity "built in" and then evaluate (perhaps with an info-theoretic style metric) the extent to which that complexity improved the model. Perhaps something similar has been done for granular materials to capture higher-order statistics of packing? I ask this primarily out of curiosity, not as a serious criticism of the current approach. A discussion of this point might add to the paper.

Thank you for these thought-provoking resources and ideas. We agree that this would be a very interesting next step to take. In particular, in Figures 1, 3, and 4 we probe cases where our predictions and observations differ. We think that future work could build upon these observations, though the framework to do so for maximum entropy packing may not yet exist. Nevertheless, it would be very interesting to use higher-order correction terms to understand exactly what kinds of new considerations are relevant for organisms that achieve various levels of morphological complexity.

We now directly address this topic in the discussion, and discuss both the suggested higher order structure paper as well as recently published work that uses topological information about the contact network of bacterial cells to uncover universal motif distributions. By combining topological and further additions to geometric models, future research may build upon the platform constructed here.

Text Additions: Inspired by this comment, we added a new paragraph (the second) to the Discussion section.

Reviewer #2:[…] While the work is very strong overall, there are a few caveats to consider, primarily concerning the simulations. Multicellularity takes many forms by many different processes among eukaryotes. While simulations do cover a range of different morphologies and developmental processes found in nature, factors not explicitly addressed such as constraint or patterning by secreted extracellular matrix, differences in cell shape, cell migration, and others can lead to different kinds of multicellular form. The extent to which potential correlations imposed by diverse morphologies might lead to deviations from theoretical maximum entropy predictions, and how robust those deviations might be to noise is not entirely clear. Additionally, the randomness strength in simulations from high to low, while reasonable, does not appear to be grounded in empirical characterization of randomness strength in developmental processes across biological systems. Ultimately, although they leave some uncertainty as to the generality of the results, these limitations do not contradict or significantly diminish the key claims of the paper.Comments for the authors:1) The simulation results are compelling but left me with some questions. To what extent do the morphologies and processes investigated by simulations address the diverse forms of multicellularity encountered across eukaryotes? To what extent does the overall shape of the multicellular structure affect the cell packing distribution (e.g. multi-lobed structure as in Zoothamnium niveum, dichotomous branching as in Dinobryon, something with an undulating boundary)? Are there any examples of simple multicellular eukaryotes that might exhibit very strongly correlated cell positions? What is known about randomness strength or precision in developmental processes in biological systems, and if anything is known, how does this compare to values in simulations? Providing a bit more contextualization or motivation for specific choices in simulations could help address these questions and would support the generality of conclusions drawn from the simulations. Although I am convinced that the results hold for a broad range of multicellular architectures and do not think that the possible existence of a few edge cases contradicts the main conclusions of the work, it is not entirely clear to me that the effects of growth morphology, connection topology, and dimensionality have been accounted for.

As this is a detailed comment with many independent questions, we interleave them and our responses below.

“To what extent do the morphologies and processes investigated by simulations address the diverse forms of multicellularity encountered across eukaryotes?”

Broadly, multicellular growth morphologies can be sorted into two general classes: intercellular bonds may be reformable, or they can be permanent (i.e., un-reformable). Our manuscript addresses both classes.

“Permanent” intercellular bonds are not reformable if broken; mother and daughter cells remain physically attached after cell division is complete. This process occurs in many clades of multicellularity, including plants, green algae, brown algae, red algae, fungi, bacteria, and in some stages of animal development. Incomplete cell separation is one of the oldest forms of multicellular assembly and one of the most successful, dominating the planet’s biomass [1]. Thus, we have primarily focused on this class of growth morphologies, which includes experiments on snowflake yeast and *Volvox carteri*, which differ in terms of dimensionality. In two simulations, we aspire to construct these experimentally observed growth morphologies. In the remaining five simulations, we seek to address a range of other multicellular morphologies. In particular, we simulate groups with reformable, sticky intercellular bonds and groups of cells that are confined by a corralling membrane. “Sticky” aggregates are broadly prevalent in nature (e.g., flocculating yeast, bacterial aggregates, or slime molds), and growth morphologies featuring maternal membranes are common as well (e.g., animals, algae).

Of course, multicellularity is rich in form and function, and we cannot capture the growth morphologies of all multicellular organisms. However, sampling a range of growth morphologies with “reformable” and “non-reformable” bonds provides at least an initial sampling of the diversity of growth morphologies that exist. We specifically do not simulate growth morphologies of confluent tissues, whether these be assembled through permanent bonds or sticky cells. As confluent tissues exhibit packing fractions of nearly 1.0, the distribution of Voronoi volumes or areas is highly impacted by the cell size distribution, which is regulated at the cellular level.

Text additions: We added the following sentences to clarify the motivation for our simulations:

“These simulations captured a couple basic properties of multicellularity: organisms may grow in both two and three dimensions, and they may assemble with two different classes of bonds: bonds that are reformable, and bonds are not reformable. […] Finally, palintomy, or growth confined inside a maternal membrane, is also common from algae to animals.”

“To what extent does the overall shape of the multicellular structure affect the cell packing distribution (e.g. multi-lobed structure as in Zoothamnium niveum, dichotomous branching as in Dinobryon, something with an undulating boundary)? Are there any examples of simple multicellular eukaryotes that might exhibit very strongly correlated cell positions?”

This is a very interesting question. What is the role played by the boundary between ‘inside’ and ‘outside’ an organism? Geometric constraints, by definition, limit how cells can be arranged, and sufficiently strong geometric constraints, i.e., constraints that precisely determine cellular positions, could dampen or eliminate fluctuations in cell packing volumes. So, how do boundary conditions impact the cell packing distribution?

First, in the case of groups with unreformable bonds, such as snowflake yeast, Zoothamnium, and Dinobryon, cells are “frozen” into place (subject to minor displacements due to, for example, mechanical deformations). However, so long as the exact location is subject to random fluctuations, each organism will have a similar but different cell spatial structure. The maximum entropy principle thus suggests that the ensemble of all organisms precisely follows the most likely cell packing distribution, even though each individual organism deviates slightly from this distribution.

Second, we would like to clarify how we define the boundary of the organism. This is a non-trivial point: where should the boundary of a multicellular organism be drawn? In some cases, for example cells contained within an epithelial sheet (such as *Volvox*), the answer is fairly clear. However, even with snowflake yeast, an algorithm must be applied to mathematically define inside versus outside. We used multiple approaches to test that our results did not depend on the method chosen. In most analyses, we used a convex hull to determine the boundary of the organism. In others, we examined “shells” within snowflake yeast clusters; we calculated the Voronoi tessellation of the entire dataset, but then separately analyzed the distribution of Voronoi volumes for cells within a range of distances from the cluster center of mass (e.g., those within 15 microns). For distances smaller than the cluster radius, the shape of Voronoi volumes is unaffected by the exact choice of boundary. Crucially, we find that both approaches exhibit maximum entropy distributions of cell packing volumes, suggesting that the convex hull method and the rough, fractal-like boundary of snowflake yeast does not significantly influence the accuracy of maximum entropy cell packing predictions.

Having clarified those points, we now more directly address the referee’s question: could a particular boundary condition make cell positions more correlated, damping fluctuations and producing deviations from maximum entropy predictions? While we cannot completely address this question, at a minimum it is known that packing statistics in thermal systems are different near and far from boundaries. This phenomenon is well demonstrated in soft matter physics, impacting everything from colloidal particles to polymers [3, 5, 7, 8, 9]. Thus, we expect that boundaries can and do impact packing statistics in multicellular organisms. In particular, it is likely that the maternal membrane we simulate causes similar effects. To test this hypothesis, we returned to our simulations of palintomy within a confining membrane to examine how the boundary impacted cell packing statistics. We bin cells into shells of finite width centered at the center of mass of the organism. As expected, we find that the distribution of Voronoi volumes depends on the distance from the boundary; cells located near the boundary are packed more densely than cells deeper in the interior. However, we find that packing within the ‘core’ and within the ‘shell’ still follow separate maximum entropy predictions (see Figure 2—figure supplement 3).

There are, in fact, a number of simple multicellular organisms with highly correlated cell positions. *Volvox carteri* is just such an example. More broadly, linear filaments (e.g., cyanobacteria) are highly correlated; the topological constraints of the bond network in linear filaments present strong non-stochastic effects, and their cell packing distribution will generally not follow the k-gamma distribution. Last, organisms that form with confluent cell layers will likely be highly correlated as the distribution of Voronoi volumes or areas will largely arise from the distribution of cell size, a trait that is highly regulated at the cellular level.

Text additions: To make this subject more clear to the reader, we have revised the first paragraph of the subsection “Snowflake Yeast” in the section “Experimental tests of multicellular maximum entropy predictions”. We have changed this paragraph to read as follows:

“Snowflake yeast grow via incomplete cytokinesis, generating branched structures in which mother-daughter cells remain attached by permanently bonded cell walls (Figure 1A). […] Conversely, if there are strong correlations in the locations of daughter cells, then we will observe deviations from maximum entropy predictions regarding the cell neighborhood volumes.”

We also added a sentence about confluent cell layers: “However, we would not expect this prediction to hold for confluent cell layers (i.e. packing fraction *φ* ≈ 1), since in general cell size is a highly regulated process; when cells tile space, the cell size distribution is highly correlated with the cell neighborhood volume distribution.”

Figure additions: We have added Figure 2—figure supplement 3, in which we show that in simulations of cells confined by a maternal membrane, the membrane poses boundary effects on the packing of the group, as shown in panel A. In panel A, we show the volume of the Voronoi volumes as a function of the distance from the center of mass (in normalized units). Then, we partition the membrane into spherical shells and show that within each shell, the distribution of Voronoi volumes is well-described by the k-gamma distribution (panels B-E).

“What is known about randomness strength or precision in developmental processes in biological systems, and if anything is known, how does this compare to values in simulations?”

A number of studies have shown that fluctuations can be crucial to highly-functioning developmental processes, for instance in sheet folding sepal growth [2, 4]. Such studies tell us three important things: (i) randomness can be just as integral a component of development as directed growth; (ii) it may be difficult to disentangle exactly which reproducible qualities and traits stem from a regulated developmental process, and which stem from random processes; and (iii) these studies imply that random processes may be more prevalent in development than we currently know, as there are not many tools by which we can actually measure the effect of randomness. With this in mind, our manuscript provides a mechanism for testing the effect of randomness on cell packing distributions. Our simulations provide test cases for what we might expect to observe in the presence or absence of randomness. We show how deviations from maximum entropy packing predictions occur, and that they indicate the presence of correlations. With this information, future work can then use this tool to quantify the randomness strength of various developmental processes.

Text additions: To highlight better this point, we have modified a paragraph of the text. The first paragraph of the section “The crucial role of randomness” now reads:

“How much randomness is necessary for the k-gamma distribution to predict cell neighborhood size distributions? […] We do not attempt to recapitulate exactly any naturally-occurring levels of randomness strength; rather, we are looking to investigate the limits about how well maximum entropy predictions can measure a deviation from pure randomness.”

2) The sections titled "Multicellular motility is robust to cellular area heterogeneity" starting on p. 11 is slightly perplexing. It is certainly interested, and I see that it addresses a question that may arise from analysis of Volvox cell packing, but in its current form, I do not believe it contributes substantially to the key points of the paper. The introduction section seemed to imply that the results would demonstrate that fluctuations in cell packing may play a role in the evolution of multicellular systems, but as I understood them, the results suggest that fluctuations do not affect motility, at least implying that there should be little to no effect on any aspect of fitness related to motility. It is possible that there could be other aspects of organismal fitness related to cell packing, so while these results are consistent with cell packing fluctuations not necessarily impeding or constraining the evolution of multicellularity, they do not strongly support that conclusion. Perhaps contextualizing the results a bit more in terms of key points of the paper while reporting them and referring to them in the Discussion section might help the reader better appreciate their significance within the context of the paper overall.

We agree that this section could be made clearer to the reader. Broadly, random perturbations could be detrimental, beneficial, or neutral to an organism. Naively, we may expect that perturbations push an organism further from an ‘ideal’ morphology and thus are largely detrimental. With snowflake yeast, we show that randomness in cell positions can lead to highly repeatable group level traits, which allows emergent multicellular traits to become remarkably heritable (Zamani-Dahaj et al., 2021: doi.org/10.1101/2021.07.19.452990). In this way, entropic cell packing imbues nascent multicellular organisms like snowflake yeast with an evolutionary benefit, allowing natural selection to act upon multicellular traits that emerge from the mechanics of cellular packing. Our analysis of *Volvox* swimming speed shows that randomness can also be neutral. One may have expected that swimming speed is optimized via precise arrangement of flagella; instead, our results show that, to leading order, swimming speed is unaffected by cell packing fluctuations. Thus, while randomness does not produce a benefit to *Volvox* swimming speed, the fact that optimal swimming does not require an ordered array of flagella represents the absence of a barrier. Swimming speed, a complex group-level trait, can be optimized without first evolving a developmental mechanism that produces precisely arranged cells.

Text additions: We make the following additions to the text to help the reader contextualize these results:

“However, the effect of heterogeneity in cell area on motility is as yet unknown; naively, we might expect that heterogeneities in flagella packing may lower the motility of the organism in comparison to a highly regular arrangement. […] The absence of such a barrier may represent a scenario where trait optimization can be achieved without first evolving sophisticated developmental genes.”

3) I might suggest removing or otherwise modifying the phrase "highly-evolved" (p.14) as its meaning is unclear, has connotations of evolutionary teleology, and clashes with the fact that all extant organisms share an evolutionary history of equal length. Maybe something such as "organisms with highly-regulated development" may be more appropriate.

We agree and have made the suggested change.

4) Is anything known about the source of correlated subregions of cells in Volvox? Do the authors have any ideas about this? Either way, it would be interesting to know and may warrant at least a small comment in the text.

We agree that this question should be addressed in the manuscript. *Volvox* exhibit a known anterior/posterior polarity. The organism will orient its anterior pole towards a light source and swim in that direction. It is possible that this polarity also affects the somatic cell area distribution.

Typically, an anterior/posterior polarity is measured by observing a moving organism: however, in our snapshot data, the organism is stationary. In other cases, the distribution of the germ cells within the interior of the organism can be used, at some stages of the *Volvox* lifespan, to correlate with the anterior/posterior positions. However, we have considered here only the somatic cells, and although we have returned to the original images and labeled the germ cells, we have found that, at best, any labeling of anterior/posterior is messy and unreliable due to uncertainty both in the position of the germ cells and in extrapolating those positions to define a definitive pole. Therefore, we do not report these measurements here, but instead discuss the idea that anterior/posterior polarity may affect the correlated subregions of the somatic cell areas.

Text additions: We have revised the first paragraph of the subsection “The role of spatial correlations” to read:

“While we have shown that the distribution of cell neighborhood volumes closely follows the k-gamma distribution in two very different organisms, we have also seen that in some cases maximum entropy predictions are more accurate in sub-sections of an organism than across its entirety. […] This polarity may affect the somatic cell packing distribution in different subregions of the organism, leading to deviations from maximum entropy predictions.”

5) In the author list, SSH is missing an asterisk to denote corresponding authorship.

Corrected.

6) An "e" is missing in "surface" in the caption for Figure 2B.

Corrected.

7) I believe the dotted red line in Figure 4B should be a solid line to match those in panels A and C.

We appreciate your close scrutiny of our figures. Here, the dotted red line signifies that there is only a small, discrete number of observed cell volumes when there is no noise added to the system. We modified the caption to emphasize this point and ensure it is clear to the reader.

Text additions: We added the following sentences to the figure caption:

“Note: the discrete points in (B) arise because, absent randomness in the cell locations, the exact same cellular structure is achieved by every simulation. […] Upon adding randomness, the cellular structure is altered between successive simulations, and the distribution again achieves continuity.”

References:

1. Yinon M. Bar-On, Rob Phillips, and Ron Milo. “The biomass distribution on Earth”. In: Proceedings of the National Academy of Sciences 115.25 (2018), pp. 6506–6511. issn: 10916490. doi: 10.1073/pnas.1711842115.

2. Pierre A. Haas et al. “The noisy basis of morphogenesis: Mechanisms and mechanics of cell sheet folding inferred from developmental variability”. In: PLoS Biology 16.7 (2018), pp. 1–37. issn: 15457885. doi: http://dx.doi.org/10.1371/journal.pbio.2005536.

3. K. Harth, A. Mauney, and R. Stannarius. “Frustrated packing of spheres in a flat container under symmetry-breaking bias”. In: Physical Review E – Statistical, Nonlinear, and Soft Matter Physics 91.3 (2015), pp. 1–5. issn: 15502376. doi: 10.1103/PhysRevE.91.030201.

4. Lilan Hong et al. “Variable Cell Growth Yields Reproducible OrganDevelopment through Spatiotemporal Averaging”. In: Developmental Cell 38.1 (2016), pp. 15–32. issn: 18781551. doi: http://dx.doi.org/ 10.1016/j.devcel.2016.06.016.

5. S´ara L´evay et al. “Frustrated packing in a granular system under geometrical confinement”. In: Soft Matter 14.3 (2018), pp. 396–404. issn: 17446848. doi: 10.1039/c7sm01900a.

6. Maciej Lisicki et al. “Swimming eukaryotic microorganisms exhibit a universal speed distribution”. In: *eLife* 8 (2019), pp. 1–20. issn: 2050084X. doi: 10.7554/*eLife*.44907.

7. Joerg Reimann et al. “Pebble bed packing in prismatic containers”. In: Fusion Engineering and Design 88.9-10 (2013), pp. 2343–2347. issn: 09203796. doi: 10.1016/j.fusengdes.2013.05.100. url: http://dx.doi.org/10.1016/j.fusengdes.2013.05.100.

8. Mohammad Mahdi Roozbahani, Bujang B.K. Huat, and Afshin Asadi. “Effect of rectangular container’s sides on porosity for equal-sized sphere packing”. In: Powder Technology 224 (2012), pp. 46–50. issn: 00325910. doi: 10.1016/j.powtec.2012.02.018. url: http://dx.doi.org/10.1016/j.powtec.2012. 02.018.

9. Yu G. Stoyan and G. N. Yaskov. “Packing identical spheres into a cylinder”. In: International Transactions in Operational Research 17.1 (2010), pp. 51–70. issn: 14753995. doi: 10.1111/j.1475-3995.2009.00733.x.